# Lamellar projections in the endolymphatic sac act as a relief valve to regulate inner ear pressure

Ian A Swinburne[1], Kishore R Mosaliganti[1], Srigokul Upadhyayula[2,3,4], Tsung-Li Liu[4], David G C Hildebrand[5], Tony Y -C Tsai[1], Anzhi Chen[1], Ebaa Al-Obeidi[1], Anna K Fass[1], Samir Malhotra[1], Florian Engert[5], Jeff W Lichtman[5], Tomas Kirchhausen[2,3,6], Eric Betzig[4], Sean G Megason[1]*

[1]Department of Systems Biology, Harvard Medical School, Boston, United States; [2]Department of Pediatrics, Harvard Medical School, Boston, United States; [3]Program in Cellular and Molecular Medicine, Boston Children's Hospital, Boston, United States; [4]Janelia Research Campus, Howard Hughes Medical Institute, Ashburn, United States; [5]Department of Molecular and Cellular Biology, Harvard University, Cambridge, United States; [6]Department of Cell Biology, Harvard Medical School, Boston, United States

**Abstract** The inner ear is a fluid-filled closed-epithelial structure whose function requires maintenance of an internal hydrostatic pressure and fluid composition. The endolymphatic sac (ES) is a dead-end epithelial tube connected to the inner ear whose function is unclear. ES defects can cause distended ear tissue, a pathology often seen in hearing and balance disorders. Using live imaging of zebrafish larvae, we reveal that the ES undergoes cycles of slow pressure-driven inflation followed by rapid deflation. Absence of these cycles in *lmx1bb* mutants leads to distended ear tissue. Using serial-section electron microscopy and adaptive optics lattice light-sheet microscopy, we find a pressure relief valve in the ES comprised of partially separated apical junctions and dynamic overlapping basal lamellae that separate under pressure to release fluid. We propose that this lmx1-dependent pressure relief valve is required to maintain fluid homeostasis in the inner ear and other fluid-filled cavities.
DOI: https://doi.org/10.7554/eLife.37131.001

*For correspondence: megason@hms.harvard.edu

**Competing interests:** The authors declare that no competing interests exist.

## Introduction

Understanding the mechanisms by which organs use water-filled cavities to compartmentalize biochemical and biophysical environments is a fundamental problem. Because water is nearly incompressible, several organs harness and cope with water as an object that transmits force. Hydrostatic pressure inflates the eye during development (*Coulombre, 1956*). Later, unstable ocular pressure from reduced production of aqueous humor or reduced drainage can lead to blindness, as occurs in hypertonic maculopathy or glaucoma (*Costa and Arcieri, 2007*; *Leske, 1983*). Hydrostatic pressure appears to drive expansion of brain ventricles during development (*Desmond, 1985*; *Desmond and Levitan, 2002*; *Lowery and Sive, 2005*). Later, unstable hydrostatic pressure in brain ventricles is correlated with hydrocephaly and mental disorders (*Hardan et al., 2001*; *Kurokawa et al., 2000*). Hydrostatic pressure inflates and controls the size of the developing ear (*Abbas and Whitfield, 2009*; *Hoijman et al., 2015*; Mosaliganti et al., unpublished). Later, unstable pressure in the ear can cause deafness and balance disorders like Pendred syndrome and Ménière's disease (*Belal and Antunez, 1980*; *Schuknecht and Gulya, 1983*). This theme of harnessing hydrostatic pressure for

**eLife digest** The most internal part of the human ear, the inner ear, is essential for us to hear and have a sense of balance. It is formed by a complex series of connected cavities filled by a liquid. When sound waves and changes in the position of the body make this liquid move, specialized 'hair' cells can detect these subtle movements; neurons then relay this information to the brain where it is decoded and interpreted.

For the inner ear to work properly, the body needs to finely regulate the pressure created by the liquid inside the cavities. For example, people with unstable pressure in their ears can experience deafness or problems with balance. A structure known as the endolymphatic sac, which is a balloon-like chamber connected to the rest of the inner ear by a thin tube, helps with this regulation. However, scientists are still unsure about how exactly the sac performs its role. One problem is that the inner ear is difficult to study because it is encased in one of the densest bones in the body.

Many other animals also have inner ears, from fish to birds and mammals. Here, Swinburne et al. examine the inner ear of zebrafish embryos because, in this fish, the ear starts working before the bones around it form; the structure is therefore accessible for injections and microscopy. Experiments show that when the pressure in the inner ear rises, the endolymphatic sac slowly fills up with the ear liquid, and then it rapidly deflates. Fish with mutations that stop the sac from deflating have overinflated sacs, which is a symptom also found in certain patients with hearing and balance disorders.

Looking into the details of these inflation-deflation cycles, Swinburne et al. found that the cells that form the sac have gaps between them, unlike a normal sheet of cells. A flap covers these gaps to keep the liquid in, but under pressure, the flap opens and the liquid can escape. These results show that the endolymphatic sac works as a pressure relief valve for the inner ear.

Ultimately, understanding how pressure is regulated in the ear could help patients with inner ear disorders. It could also serve as a template to investigate how eyes, kidneys and the brain, which all have liquid-filled cavities, control their internal pressure.

DOI: https://doi.org/10.7554/eLife.37131.002

normal development and controlling pressure for healthy physiology raises the question of how tissues regulate pressure. Tissue structures identified as important for pressure control include Schlemm's canal in the eye, arachnoid granules and the choroid plexus in the brain, and the endolymphatic duct and sac in the ear (*Johnstone et al., 2011*; *Kimura and Schuknecht, 1965*; *Naito, 1950*; *Orešković et al., 2017*; *Pollay, 2010*; *Symss and Oi, 2013*; *Tripathi, 1974*). A cavity's pressure could be managed via mechanisms involving molecular pores, molecular transporters, and the physical behavior of the tissue. Observing the mechanisms by which these tissue barriers control pressure has been limited by a range of obstacles such as optical accessibility and uncertainty in both the time- and length-scale on which they function. As such, not much is known about how these tissues manage fluctuating pressures because they have not been observed in vivo.

The inner ear is a prominent example of an organ whose tissue form determines its physiology. It is filled with endolymph, the composition of which differs from other fluids such as plasma, perilymph, and cerebral spinal fluid in that it contains high potassium, low sodium, and high electric potential (*Lang et al., 2007*). This endolymph composition is necessary to drive ion currents into hair cells to convert fluid movement, caused by either the head's acceleration or by sound, into biochemical signals that underlie balance and hearing. The ear's endolymphatic duct connects the endolymph in the semicircular canals, cochlea, and other chambers to the endolymphatic sac (ES, *Figure 1A,B*). An ES-like structure is present in basal vertebrates, including lamprey and hagfish (*Hammond and Whitfield, 2006*), suggesting it plays an ancient role in inner ear function. The epithelium of the ES, as well as that of the rest of the ear, has an apical surface facing the internal endolymph and a basal surface facing the external perilymph (*Figure 1A*). Most of the adult ear is enclosed in a bony labyrinth within the temporal bone and perilymph is located in the space between the epithelium and bone (*Figure 1A*). In adults, the endolymphatic duct and sac are partially encased within the temporal bone such that the distal end of the endolymphatic sac protrudes out of the temporal bone into the cranial cavity (*Figure 1A*). Excessive hydrostatic pressure can tear

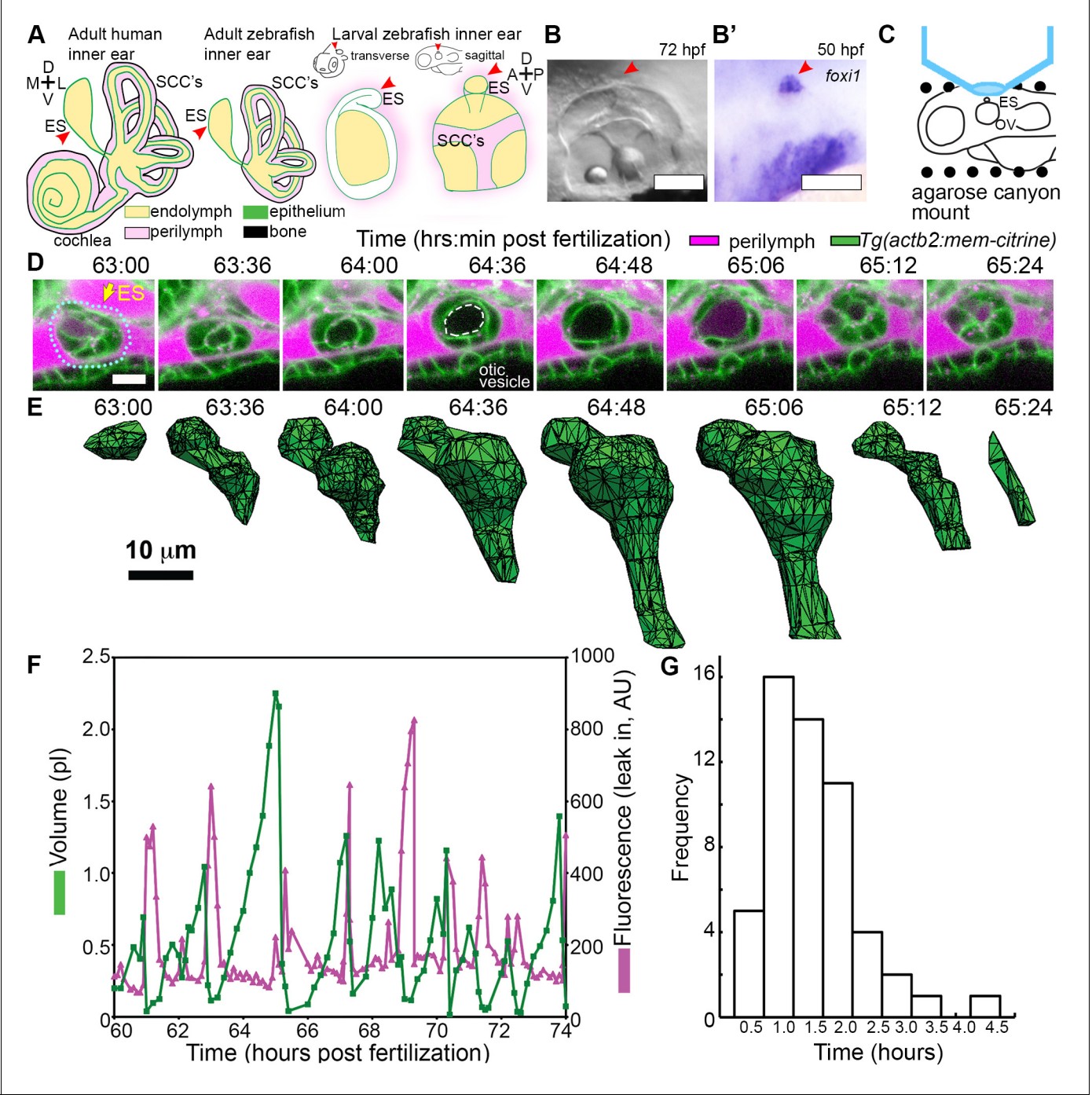

**Figure 1.** ES lumen slowly inflates and rapidly deflates every 0.3–4.5 hr. (**A**) Illustration of the adult human inner ear showing cochlea, semicircular canals (SCCs), and endolymphatic duct and sac (ES, red arrowhead) and their organization of tissue (green), endolymph (beige), perilymph (magenta), and bone (black). Illustrations of the adult and larval zebrafish inner ear showing ES (indicated with red arrowheads, see also *Figure 1—figure supplement 1* and *Video 1* for how the zebrafish ES first forms). (**B**) Micrograph of larval zebrafish, sagittal view. (**B'**) In situ of *foxi1* highlights position of ES (red arrowhead), *n* = 12. (**C**) Illustration of imaging setup. (**D**) Slices and select time points from 3D confocal time course showing a single inflation and deflation event from a live zebrafish embryo. Cell membranes (green) are labeled using ubiquitous membrane citrine transgenes. Perilymph (magenta) is labeled with 3 kDa dextran-Texas red. ES identified with dotted blue outline and yellow arrow. Lumen of inflated ES identified with dashed white outline in 64:36 panel. (**E**) Corresponding 3D meshes of the segmented ES lumen volume. (**F**) Quantification of segmented ES volumes (primary axis, green) and leak in fluorescence (secondary axis, magenta) over multiple cycles (see also *Figure 1—figure supplement 1B,C* and *Videos 2–3*). (**G**) Histogram of times between peak inflation volumes, compiled from eight different time courses and 54 inflations. Scale bars 100 µm for (**B**) and 10 µm for (**D,E**).

*Figure 1 continued on next page*

*Figure 1 continued*

DOI: https://doi.org/10.7554/eLife.37131.003

The following figure supplement is available for figure 1:

**Figure supplement 1.** Early ES development and additional examples of wild-type ES inflation and deflation.

DOI: https://doi.org/10.7554/eLife.37131.004

the inner ear's epithelium, disrupting its electric potential (a pathology called endolymphatic hydrops). In contrast, low potassium or reduced endolymph production can lower hydrostatic pressure within the ear to the point where the ear chambers may collapse (*Lang et al., 2007*). Early work showed that ES ablation causes hydrostatic pressure to rise and the epithelium to tear (*Kimura and Schuknecht, 1965*; *Naito, 1950*), suggesting that the ES may maintain endolymph homeostasis. Recent work indicated that an explanted mouse ES gradually loses endolymph (*Honda et al., 2017*). While many molecular pores and transporters are expressed in the ES, it is unclear why it is organized as a dead-end tube and how it might reduce endolymph volume.

The ES has not been studied much in zebrafish because it has not always been clear that zebrafish possess this structure or at what time it forms during development (*Haddon and Lewis, 1996*; *Hammond and Whitfield, 2006*). In situ hybridization images at 52 hr post fertilization (hpf) showing expression of ES markers, *foxi1* and *bmp4*, restricted to an ES-like structure positioned on the dorsal side of the otic vesicle are the strongest and developmentally earliest evidence for zebrafish having an ES (*Figure 1B'*) (*Abbas and Whitfield, 2009*; *Geng et al., 2013*). However, its formation and physiological function remain unknown. Here, we present a detailed characterization of the ES in zebrafish performed using confocal microscopy, serial-section electron microscopy, adaptive optics lattice light-sheet microscopy, and a genetic mutant that together demonstrate that the ES contains a physical relief valve for regulating inner

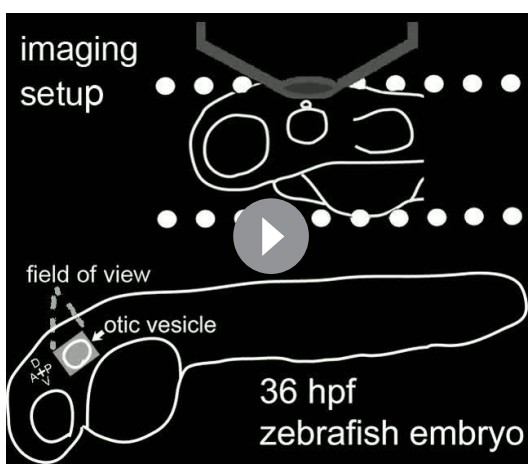

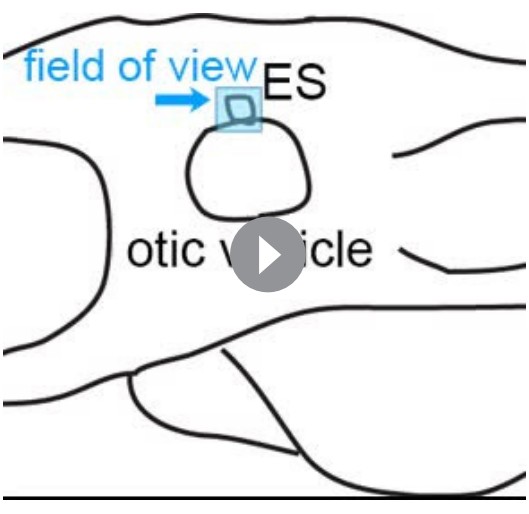

**Video 1.** Early ES development. Video begins with schematic of experimental set-up and context of the presented field of view. Then, an annotated time point is presented of the upcoming video. The presented video is of a sagittal slice from a 4D time course of early ES development (white arrow points to ES in introduction). ES morphogenesis begins at 36 hr post fertilization (hpf) as an evagination in the dorsal-anterior-lateral epithelial wall of the otic vesicle. Fluorescence from membrane citrine, shown in grey. Scale bar is 10 μm.

DOI: https://doi.org/10.7554/eLife.37131.005

**Video 2.** Wild-type ES inflates and deflates. Video begins with an illustration depicting the context of the presented field of view, which is a sagittal slice encompassing the developing ES from a 4D time course. Then, an annotated time point is presented of the upcoming video, a green arrow points to the ES, a dotted line outlines the ES lumen, the otic vesicle is labeled ventral to the ES, and the perilymph surrounds the ES structure, labeled in magenta. Video of sagittal slice from 4D dataset, quantified in *Figure 1F*. Fluorescence from membrane citrine, shown in green. Perilymph highlighted with fluorescence from 3 kDa dextran-Texas red, shown in magenta. Scale bar is 10 μm.

DOI: https://doi.org/10.7554/eLife.37131.006

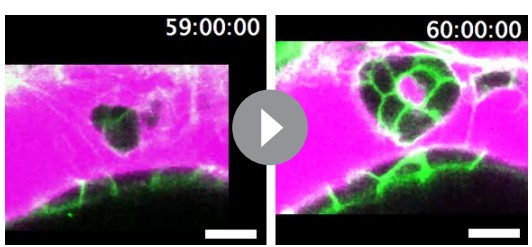

**Video 3.** Wild-type ES inflates and deflates. A video of two time courses, sagittal slices from 4D datasets, quantified in *Figure 1—figure supplement 1B* (left) and *Figure 1—figure supplement 1C* (right). Fluorescence from membrane citrine, shown in green. Perilymph highlighted with fluorescence from 3 kDa dextran-Texas red, shown in magenta. Scale bars are 10 µm.

DOI: https://doi.org/10.7554/eLife.37131.007

ear fluid pressure through regulated transepithelial fluid release.

## Results

### The endolymphatic sac exhibits cycles of inflation and deflation

We first established an imaging system to identify the developmental origins of the zebrafish ES. Extended time-lapse imaging required long-term immobilization with α-bungarotoxin that permits normal development without the reduction in ear growth caused by long-term treatment with tricaine, bright fluorescent transgenic fish with contrast from a membrane-localized fluorescent protein to minimize bleaching and photoxicity, and mounting in a submerged agarose canyon that permits positioning of the zebrafish ear close to the coverslip (400 µm wide, walls of canyon secure the yolk and head, *Figure 1C*) (*Swinburne et al., 2015*). The zebrafish ear develops from a collection of epithelial cells that form a closed fluid-filled ellipsoidal structure called the otic vesicle (*Whitfield, 2015*). We saw that ES morphogenesis begins at 36 hpf as an evagination in the dorsal epithelial wall of the otic vesicle (*Figure 1—figure supplement 1A*, *Video 1*, *Figure 1A–B*, and ES identity confirmed by expression of *foxi1*, *Figure 1B'*). Between 36 and 60 hpf, the ES grows and elongates. These findings established that the zebrafish ES is optically accessible during embryonic and larval stages and that ES morphogenesis begins at 36 hpf.

The zebrafish ear starts to sense body acceleration for the nervous system between 60 and 72 hpf, as demonstrated by the onset of the vestibulo-ocular reflex (*Mo et al., 2010*). We found that the ES begins to exhibit a physical behavior during the same time window. We observed that the lumen of the ES remains closed until 60 hpf, but between 60 and 65 hpf it begins cycles of slowly inflating and rapidly deflating (sagittal slices, *Figure 1D*, lumen volumes, *Figure 1E*, *Figure 1—figure supplement 1*, and *Videos 2–3*). Three-dimensional (3D) measurements showed that the ES lumen volume changes 5–20-fold through the course of each cycle (*Figure 1E,F*, and *Figure 1—figure supplement 1B–C*). The period between peak ES volumes exhibited a broad distribution (0.3–4.5 hr) with an average of 1.6 ± 0.8 hr (mean ± SD, histogram compiled from eight time courses and 54 peaks, *Figure 1G*). These initial observations suggested several potential causes of the inflation-deflation cycles of the ES including: a response to organ-wide increases in fluid pressure within the otic vesicle, a local tissue behavior wherein the ES inflates with fluid from the perilymph, or a local tissue behavior wherein cells in the ES periodically coordinate their movements to expand the ES volume by sucking fluid from the otic vesicle.

### Loss of the epithelial barrier is sufficient for ES deflation

To assess whether ES inflations occur by transmission of endolymph pressure between the otic vesicle and the ES, we performed a single injection of a small volume of solution containing 3 kDa fluorescent dextran into the otic vesicle and followed its movement during inflation and deflation. We found that the duct is open and that endolymph flows from the otic vesicle into the ES during inflation (*Figure 2A–D*, *Video 4*). Additionally, upon deflation, endolymph rapidly leaked out of the ES and into the perilymphatic space (75:22 panel, *Figure 2D*). To determine whether pressure in the otic vesicle is transmitted to the ES for its inflation, we laser-ablated 2–3 cells within the wall of the otic vesicle distant from the ES at 64 hpf (*Figure 2E*). Shortly after ablation, the ES deflated and completely collapsed within 20 min (*Figure 2F*). These results indicate that fluid pressure in the otic vesicle inflates the ES volume, the ES tissue has elastic material properties, and a loss of epithelial integrity is sufficient for ES deflation.

A core function of epithelial sheets is to act as a barrier that can prevent passive movement of molecules between an organ's interior and exterior. Deflations could be driven by local breaks in the

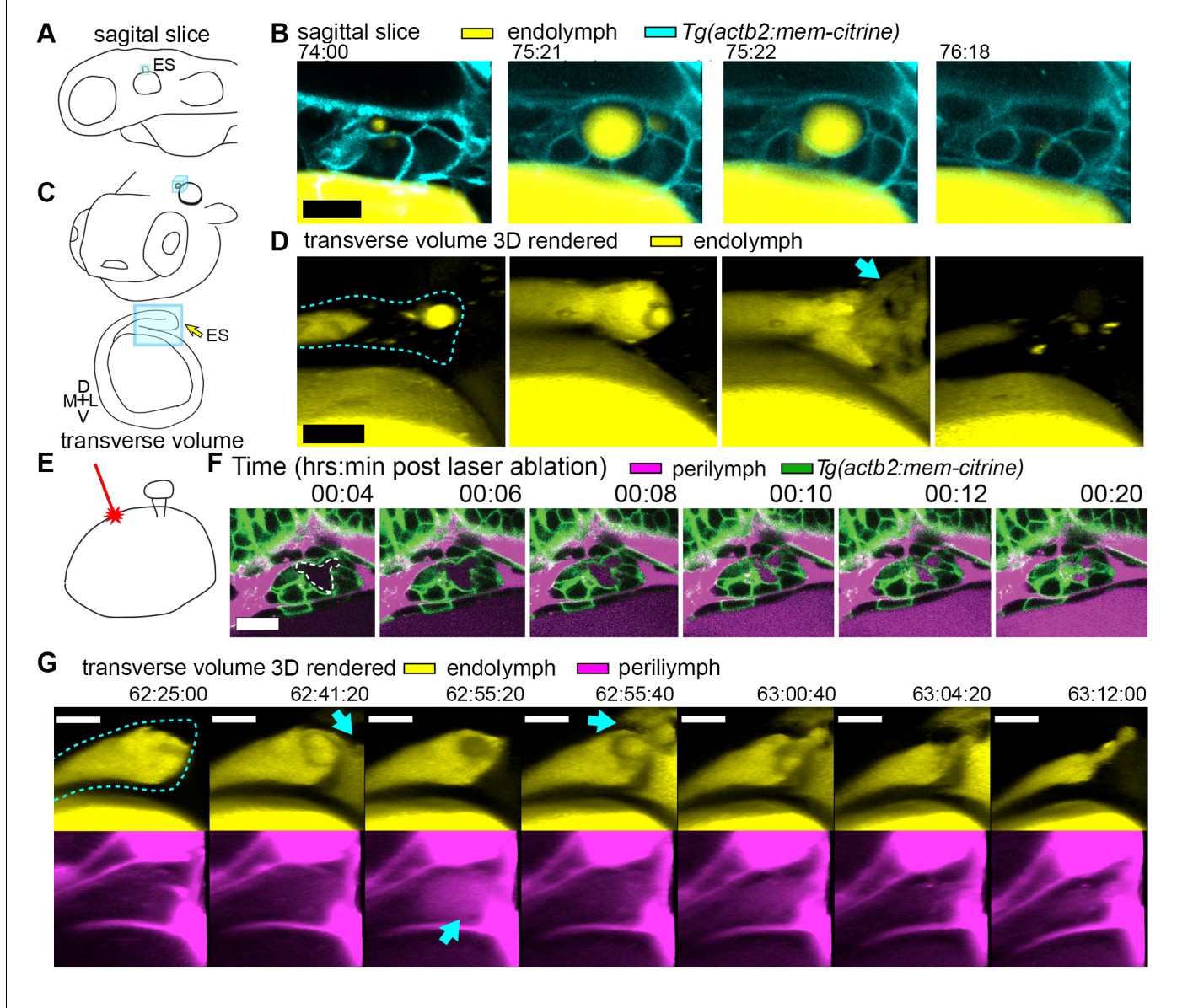

**Figure 2.** Hydrostatic pressure transmits endolymph through duct to inflate the ES. (A) Illustration of larval zebrafish highlighting sagittal plane of image acquisition (blue square). (B) Time points of individual sagittal slices of raw data from 3D time course (endolymph labeled yellow by single dye injection into otic vesicle, membrane citrine in cyan). (C) Illustration of larval zebrafish highlighting transverse perspective (blue box) for rendered volumes of ES. (D) Time points from time course of raw data rendered in 3D transverse view (endolymph in yellow, ES tissue outlined with dashed line) showing endolymph flowing through duct to ES and then out to perilymph (blue arrow). (E) Illustration of strategy for laser ablating otic vesicle cells with point-scanning 2-photon laser to ablate 2–3 targeted cells. (F) Slices from 4D confocal time course after laser ablation showing ES deflation. ES lumen is outlined with a white dashed line in (F), n = 4. (G) Time points from time course rendered in 3D transverse view (endolymph in yellow, perilymph in magenta, ES tissue outlined with dashed line), n = 8. Blue arrows indicate endolymph expulsion or perilymph leak in events (D,G). All scale bars 10 μm.
DOI: https://doi.org/10.7554/eLife.37131.008

epithelial barrier combined with elastic tissue collapse or by an alternative mechanism such as cellular reabsorption of endolymph. To distinguish between these mechanisms, we used dye injections into the heart to label the perilymph that surrounds the otic vesicle and ES (*Figure 1D*). Quantification of fluorescence within the lumen of the ES revealed that perilymph dye begins to leak into the ES lumen at the onset of each deflation (*Figure 1F*, visible leakage in *Figure 1D*, *Videos 2–3*). In the example shown deflation coincided with an influx of fluorescence from the perilymph into the lumen of the ES in each of the nine major inflation-deflation cycles. This coincident timing suggests that

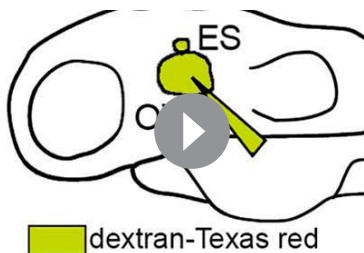

**Video 4.** Endolymph periodically inflates ES and then released into periotic space. Time course of otic vesicle injected with 3 kDa dextran-Texas red at 55 hpf. Panels are transverse volumes of same time course. Left, labeled endolymph presented in yellow. Right, labeled endolymph in yellow, membrane citrine in cyan. Scale bar is 10 μm.

DOI: https://doi.org/10.7554/eLife.37131.009

both deflation and dye leakage were due to transient breaks in the epithelial barrier, an interpretation that is consistent with the rapid release of endolymph prior to deflation (*Figure 2D*, *Video 4*).

The observation that perilymph enters the deflating ES may seem counterintuitive, since the contents of a punctured high-pressure elastic vessel might be expected to primarily flow outwards. Estimation of the rates of advection and diffusion supports the presence of upstream movement of perilymph by diffusion. Due to limited spatial resolution and the absence of contrast for trafficking vesicles, we could not dismiss alternative mechanisms such as contributions from rapid transcytosis or cellular absorption. To determine whether endolymph efflux occurs at the same time as perilymph influx, we simultaneously labeled the two fluids with different colored dyes and imaged their localization with higher temporal resolution (*Figure 2G*, *Video 5*). We found a fluttering behavior underlying deflation in which endolymph leaked out (62:41:20), perilymph leaked in (62:55:20), leaked-in perilymph flushed out with more endolymph efflux (62:55:40), followed by complete deflation (63:12:00). It is difficult to explain the simultaneous inward and outward movement of fluid with the alternative interpretation that rapid transcytosis or cellular absorption processes deflate the ES. A plausible mechanism for this observed behavior involves a structure that normally resists flow driven by a pressure differential that rapidly opens during deflation to allow for fast efflux by advection followed by influx by diffusion before re-closing.

## Lmx1bb is essential for ES deflation

The uniqueness of the ES inflation and deflation cycles suggests specific genes might be involved.

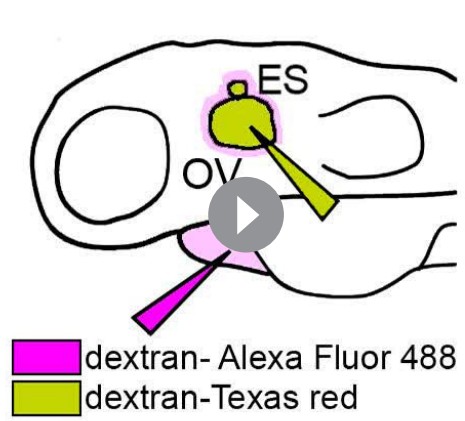

**Video 5.** Endolymph released into periotic space followed by perilymph leak-in. Time course of otic vesicle injected with 3 kDa dextran-Texas red at 55 hpf, and perilymph labeled with 10 kDa dextran- Alexa Fluor 488. First three panels are transverse volumes while the fourth is a dorsal view of the same time course. Labeled endolymph presented in yellow, perilymph in magenta. Scale bar is 10 μm.

DOI: https://doi.org/10.7554/eLife.37131.010

To identify pathways that contribute to the emergence of this physiology and to reveal ways in which it can malfunction, we examined a mutant caused by a premature stop codon in the transcription factor *lmx1bb* (*Obholzer et al., 2012*) that exhibited an enlarged ES. We found that the ES in *lmx1bb* mutants became greatly enlarged (>4 times the inflated wild-type ES volume) making it readily visible at 80 hpf by bright-field microscopy (*Figure 3A*). To determine if *lmx1bb* is expressed at an appropriate place and time for a mutation to be causing an ES defect, we imaged a transgenic reporter line, *Tg(lmx1bb: eGFP)^{mw10}*, driven by the *lmx1bb* promoter *McMahon et al., 2009*.This reporter was expressed in ES cells beginning at 52–58 hpf, just before the first ES inflation (cyan, *Figure 3B*), consistent with *lmx1bb* expression being instrumental for development of the ability of the ES to release pressure. Earlier in the development of the otic vesicle, *lmx1bb* is expressed in portions of the nascent semicircular canals and sensory patches, regions of the inner ear that also exhibit abnormal development in the mutant (*Obholzer et al., 2012*; *Schibler and Malicki, 2007*). There is no precedent for ES development

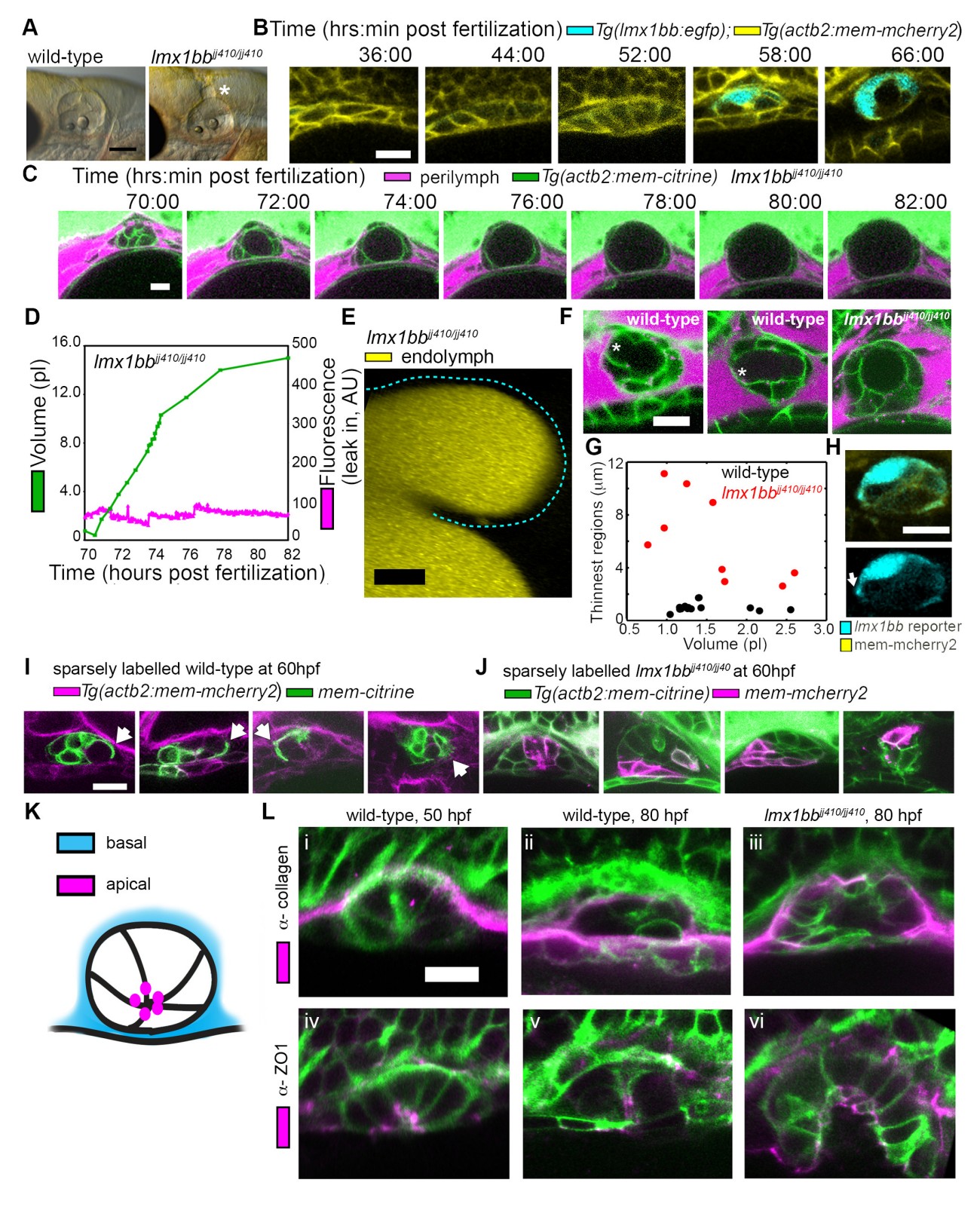

**Figure 3.** Lmx1bb is necessary for development of the ES's ability to form breaks in its diffusion barrier and deflate. (**A**) Lateral view of wild-type and *lmx1bb*^ji410/jj410^ mutant ears imaged by bright-field microscopy at 80 hpf, asterisk labels greatly enlarged mutant ES. Scale bar, 100 μm. (**B**) Slices from 3D confocal time course of an *lmx1bb* transcriptional reporter (cyan, *Tg(lmx1bb:egfp)*^mw10/mw10^; yellow, *Tg(actb2:mem-mcherry2)*^hm29^), n = 3. (**C**) Slices and select time points from 3D confocal time course of *lmx1bb*^ji410/jj410^ mutant embryos. Membrane (green) from ubiquitous membrane citrine

*Figure 3 continued on next page*

*Figure 3 continued*

transgenes. Perilymph (magenta) from 3 kDa dextran-Texas red, *n* = 4. (D) Quantification of segmented ES volumes (primary axis, green) and leak-in fluorescence (secondary axis, magenta) from *lmx1bb^{jj410/jj410}* time course in (C) (see also *Figure 3—figure supplement 1* and *Videos 5–6*). (E) 3D transverse view (endolymph in yellow) from timelapse showing endolymph in dilated mutant ES, outlined with dashed blue line, *n* = 2. (F) Small regions with thin membranes (asterisks) form in the inflated ES of wild-type but not *lmx1bb* mutants. (G) Quantification of minimum epithelial thickness versus inflated ES volume in mutant (plotted in red, *n* = 9) and wild-type (plotted in black, *n* = 14). Compiled from 65 to 80 hpf embryos. (H) Uneven labeling from *Tg(lmx1bb:egfp)* reveals thin basal processes (white arrow). (I) Wild-type ES examples with sparsely labeled cells: membrane-labeled citrine (green) in a membrane-labeled cherry background (magenta), white arrows indicate lamellar projections, *n* = 15. (J) *lmx1bb^{jj410/jj410}* mutant ES examples with sparsely labeled cells: membrane cherry (magenta) in a membrane citrine background (green), *n* = 9. (K) Cartoon schematic of apico-basal organization of ES. (L) Supporting whole-mount immuno-stains for basal and apical markers (collagen and ZO-1, both magenta) in a membrane-labeled citrine background (green), *n* = 16, 15, 6, 32, 21, and 12 for (i-vi). Scale bars in (B-L), 10 μm.

DOI: https://doi.org/10.7554/eLife.37131.011

The following figure supplement is available for figure 3:

**Figure supplement 1.** Inflation of additional mutant ES.

DOI: https://doi.org/10.7554/eLife.37131.012

being dependent on those portions of the otic vesicles and there are many mutants with similar SCC or sensory defects that do not have ES phenotypes (*Fekete, 1999*; *Malicki et al., 1996*; *Whitfield et al., 1996*). Live imaging and perilymph tracking in *lmx1bb^{jj410/jj410}* mutant embryos revealed that the ES lumen over-inflates (*Figure 3C–D*, *Figure 3—figure supplement 1*, and *Videos 6–7*). As in the wild-type analysis, we quantified the presence of perilymph leaking into the ES lumen (secondary axes of *Figure 3D* and *Figure 3—figure supplement 1A*). In the mutant, however, we never observed perilymph entering the ES. Additionally, we imaged mutants where the endolymph was labeled and did not observe leakage out of the distended ES epithelium (*Figure 3E*). These findings suggest that the epithelial diffusion barrier remains intact in the mutant ES.

## Lamellar barriers appear to form an ES relief valve

The absence of ES deflation in the *lmx1bb* mutant suggests that a structural deficiency may be present. Indeed, by comparing the dorsal ES tissue between wild-type and *lmx1bb* mutants using confocal microscopy and membrane-localized fluorescent proteins, we observed that the ES tissue was much thinner in wild-type embryos during peak inflation than in the corresponding region in mutants inflated to a similar volume (thin region indicated by asterisk, *Figure 3F,G*, data in G compiled from 65 to 80 hpf inflation events). In wild-type embryos, this region often appeared as a thin sheet (1.0 ± 0.3 μm, mean ±SD) rather than thick as seen in the un-inflated wild-type ES or the *lmx1bb* mutants in which two distinct surfaces (apical and basal) were observed (6.2 ± 3.2 μm, the mutant, *n* = 9, is significantly thicker than wild-type, *n* = 14, with a Mann-Whitney-Wilcoxon one tailed *p*-value of $4 \times 10^{-5}$). Imaging the uneven signal of the *Tg(lmx1bb:eGFP)* reporter revealed thin protrusions that extend along the basal side of neighboring ES cells (white arrow, *Figure 3H*). Sparse mosaic labeling of cells in the wild-type further confirmed the presence of thin basal protrusions in the ES, as well as diversity in their organization (*Figure 3I*). Similar labeling in the mutant revealed the absence of thin basal protrusions in the ES (*Figure 3J*). To determine the gross organization of the ES tissue, we stained for basal proteins collagen (*Figure 3K–L*) and laminin (not shown). We found that the basal surface of the ES tissue faces the perilymph

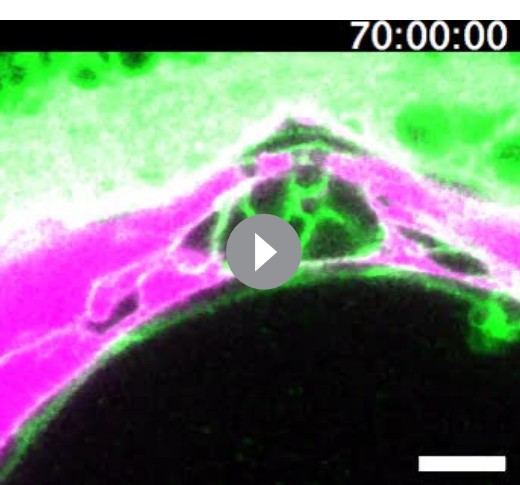

**Video 6.** Mutant ES over-inflates. Video of sagittal slice from 4D dataset of *lmx1bb^{jj410/jj410}* mutant- quantified in *Figure 3D*. Fluorescence from membrane citrine shown in green. Perilymph highlighted with fluorescence from 3 kDa dextran-Texas red, shown in magenta. Scale bar is 10 μm.

DOI: https://doi.org/10.7554/eLife.37131.013

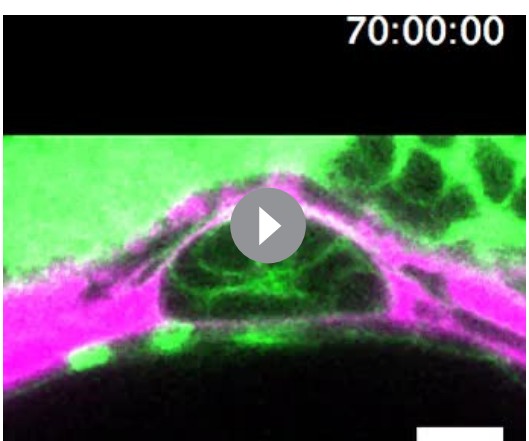

**Video 7.** Mutant ES over-inflates. Video of sagittal slice from 4D dataset of *lmx1bb^{ji410/jj410}* mutant- quantified in *Figure 3—figure supplement 1*. Fluorescence from membrane citrine shown in green. Perilymph highlighted with fluorescence from 3 kDa dextran-Texas red, shown in magenta. Scale bar is 10 µm. DOI: https://doi.org/10.7554/eLife.37131.014

outside surrounding the otic vesicle (*Figure 3L*). We could not resolve a clear difference in the staining pattern of collagen between wild-type and the *lmx1bb^{410/410}* mutant. Prior work suggested that *Col1a2* may be a direct target of LMX1B in the limb (*Haro et al., 2017*) and additional studies will be necessary to determine whether collagen or other extracellular matrix components are abnormally expressed in the ES of *lmx1bb^{410/410}* mutants. We also examined localization of the apical marker ZO-1 and found that it stained the ES tissue that faces the endolymph-filled lumen (*Figure 3K–L*). This apical-basal organization was also present in the *lmx1bb^{410/410}* mutant (*Figure 3K–L*). The absence of local breaks in the mutant ES, the absence of regional thinning in the mutant inflated ES, and the absence of thin basal protrusions suggests that these thin areas may be structurally relevant to ES pressure relief.

To investigate this possibility, we examined the ultrastructure of these thin areas using serial-section electron microscopy (EM) at a resolution of 4.0 × 4.0 × 60.0 nm per voxel. We found the dorsal tissue of the inflated wild-type sac consists of an extremely thin shell of broadly overlapping lamellae extending from the basal sides of multiple adjacent cells (*Figure 4*; *Video 8*). These basal extensions appear to be the distal barrier in the ES because the endolymph they contain is continuous with the lumen of the ES and endolymphatic duct (extending ventrally from outlined ES lumen in third panel of *Figure 4A*). We term these 'lamellar barriers' because of their thin, plate-like structure and apparent function as a barrier that holds the elevated hydrostatic pressure of the endolymph. We observed lamellar sheets that extend for distances as long as a typical cell body but are as thin as 40 nm (*Figure 4A,F*; lamella from cell segmented in purple extends 7.5 µm in the *x-y* plane and 6.6 µm along the *z*-axis). In contrast to junctions between thin aveolar cells, which kiss at their tips with tight junctions, the lamellae from ES cells formed large zones of overlap and sometimes bifurcated to form a tongue-and-groove structure (*Figure 4B*, inset) or interwoven structures (*Figure 4C, F*, black mesh highlights area of overlap, dotted perimeter indicates full spread). These structures lacked electron-dense signal, indicating that they were unlikely to contain tight junctions between lamellae. We also identified an ear where the lamellar barriers were separated as if forced into an open configuration (*Figure 4D*). This may be an ES in the act of deflating via bursting or sliding of the lamellae. The full serial-section EM dataset also confirmed that the endolymphatic duct connects the lumen of the otic vesicle to the tip of the ES, where basal lamellae form a complete barrier (available at http://zebrafish.link/hildebrand16, ES-specific links presented in methods) (*Hildebrand et al., 2017*). Basal lamellae were present along the length of the endolymphatic duct. However, cell bodies in the duct remain tightly packed with apical junctions (*Figure 4A,E*) unlike the apical and lateral membrane separations exposing lamellae in the ES.

We identified electron-dense tight junctions between cells at the tip of the ES. These same cells also had lamellar projections (yellow arrows indicate tight junctions, *Figures 4G* and *3* serial EM sections, 1.2 µm apart, the diameter of the apical opening is ~1.2 µm). The electron-dense signal formed a ring sealing the apical side of these ES cells that was continuous except for an opening that connected fluid from the duct and sac to the exposed lamellae (magenta arrow directed from duct to sac, *Figure 4G*). Apical junctions were also present in the mutant ES of *lmx1bb^{ji410/jj410}* larvae (yellow arrows, *Figure 4H,H'*). Unlike the wild-type ES, however, the mutant ES appeared to completely lack openings in its apical junctions, as we were unable to identify any in mutants imaged by serial-section scanning EM or transmission EM (four total mutant ears). Consistent with the absence of thin protrusions in the sparsely labeled mutants (*Figure 3J*), we could not find long lamellar projections on the basal side of the mutant ES (*Figure 4I*). We were unable to identify apical

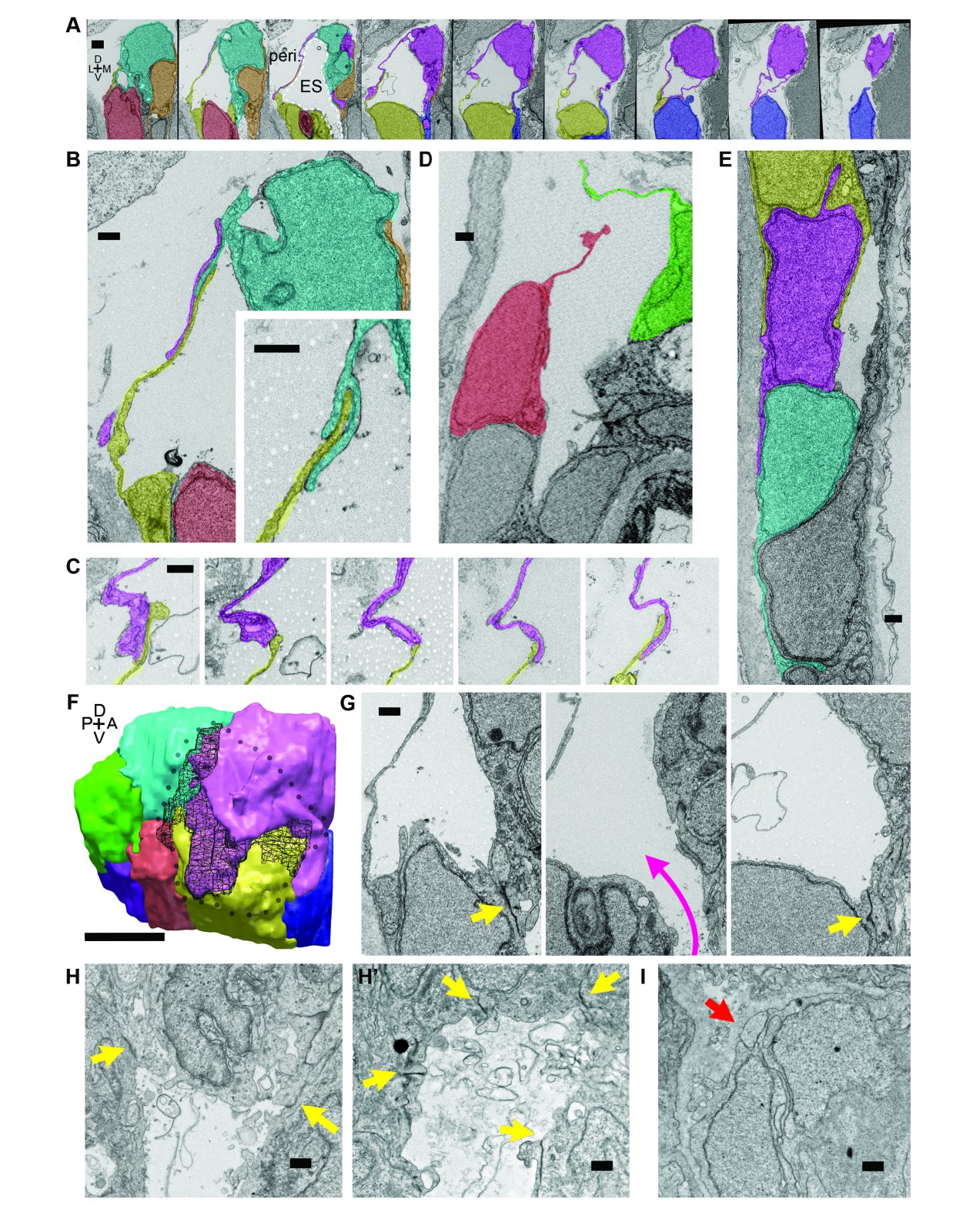

**Figure 4.** Lamellar protrusions at the tip of the ES exist in open and closed configurations. (**A**) Select images from serial-section scanning electron microscopy of a 5.5 dpf zebrafish's right inner ear. Dorsal is up, lateral left, medial right, ventral down, anterior top, and posterior bottom of the series z-stack. Cells forming lamellar barriers were labeled (color overlays, consistent across panels) to highlight connectivity of lamellae. Presented slices are a subset from the series, each separated by 960 nm (**Video 8**). The lumen of the ES is labeled and outlined with a white dashed line in third panel;
*Figure 4 continued on next page*

*Figure 4 continued*

perilymph (peri.). (**B**) Lamellae interdigitate and can form tongue-in-groove structures (inset). (**C**) Lamellae can interweave. (**D**) Example of lamellar junctions in an open configuration. (**E**) Cells in endolymphatic duct have basal lamellae, with the presented duct connecting with ES in panel A. (**F**) 3D rendering of ES segmentation from serial micrographs shown in panel (**A**) and *Video 8*. Black dotted-outline encompasses area of closed, endolymph-filled lamellae. Black mesh highlights areas of membrane overlap between lamellae that are spread open. (**G**) Electron-dense tight junctions (yellow arrows) present in cells that also have spread basal protrusions. An opening in the apical junctions creates a path from the duct to the basal protrusions (magenta arrow, slices in three panels 1.2 μm apart). (**H, H'**) *lmx1bb^{jj410/jj410}* embryos maintain apical junctions between ES cells (yellow arrows). (**I**) Mutant ES cells lack basal protrusions (red arrow). Scale bar in (**A**) is 1000 nm, (**F**) is 5 μm, and all other scale bars are 500 nm. (**B–G**) Serial section electron microscopy, *n* = 2, additional transmission EM, *n* = 3. (**H–I**) Serial section EM of mutant, *n* = 1, transmission EM, *n* = 3.

DOI: https://doi.org/10.7554/eLife.37131.015

openings in the wild-type ZO-1 immunostains (*Figure 3L*). This could be due to difficulty resolving small openings (~1 μm diameter) or to ZO-1 remaining present at the openings such that the apical barrier could reform after ES deflation to hold hydrostatic pressure. These data suggest that *lmx1bb*-dependent activity is necessary for openings to form in the apical junctions of ES tissue and for ES cells to extend thin basal protrusions.

## Cells stretch during ES inflations

The stress from increased pressure in the otic vesicle likely causes an increased ES volume by inducing cell stretching or expansion of the lamellar barrier. To distinguish how the tissue behaves through cycles of ES inflation and deflation, we sought better resolution during live imaging using lattice light-sheet microscopy (LLSM), which generates thin light-sheets using Bessel beams to enhance axial resolution while minimizing phototoxicity and bleaching (*Chen et al., 2014*; *Gao et al., 2014*). When using LLSM to image the ES, emitted light passes through brain tissue that scatters and refracts light in a complex, spatially uneven manner. Recent advances in the application of adaptive optics (AO) to microscopy can compensate for these aberrations (*Wang et al., 2014*). A microscope combining live-cell lattice light-sheets and adaptive optics was built (*Liu et al., 2018*) and used here to image ES cycles with significantly improved spatial and temporal resolution and reduced photo-damage owing to 2–4 times less laser power being distributed over a large volume (the imaging plane is volumetrically ~$10^5$ times larger than the confocal point).

LLSM requires the illumination plane and optical axis to be perpendicular to one another, so we developed a mold for an agarose mount shaped like a volcano that positions the embryo in the desired orientation (*Figure 5A*). The LLSM produced slightly blurred images with low signal-to-noise (SNR, *Figure 5B*) without adaptive optics correction. Adding the correction with an AO-LLSM system adequately compensated for tissue aberrations and produced images with higher SNR and better contrast. The resulting high-quality images were then analyzed using our software, ACME, to reconstruct membrane signals (*Figure 5C*), segment cells from the lumen of the ES (*Figure 5D,E*), and track cells and lumen accurately through time (*Mosaliganti et al., 2012*). As with confocal live imaging, we observed cycles of the ES lumen inflating and deflating (*Figure 5F,G*; *Videos 9–10*).

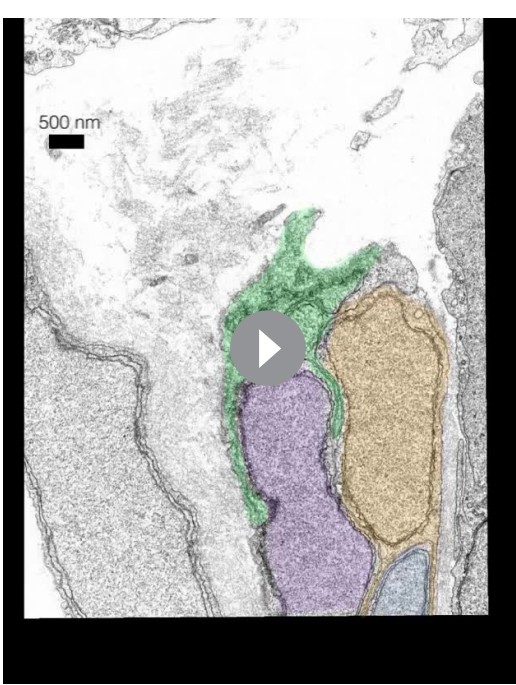

**Video 8.** Serial-section electron micrographs of wild-type ES at 5.5 dpf. Sections are 60 nm thick and color overlays highlight cells with lamellar barriers or basal lamellae. Second half explores organization of cells in space using cell segmentations.

DOI: https://doi.org/10.7554/eLife.37131.016

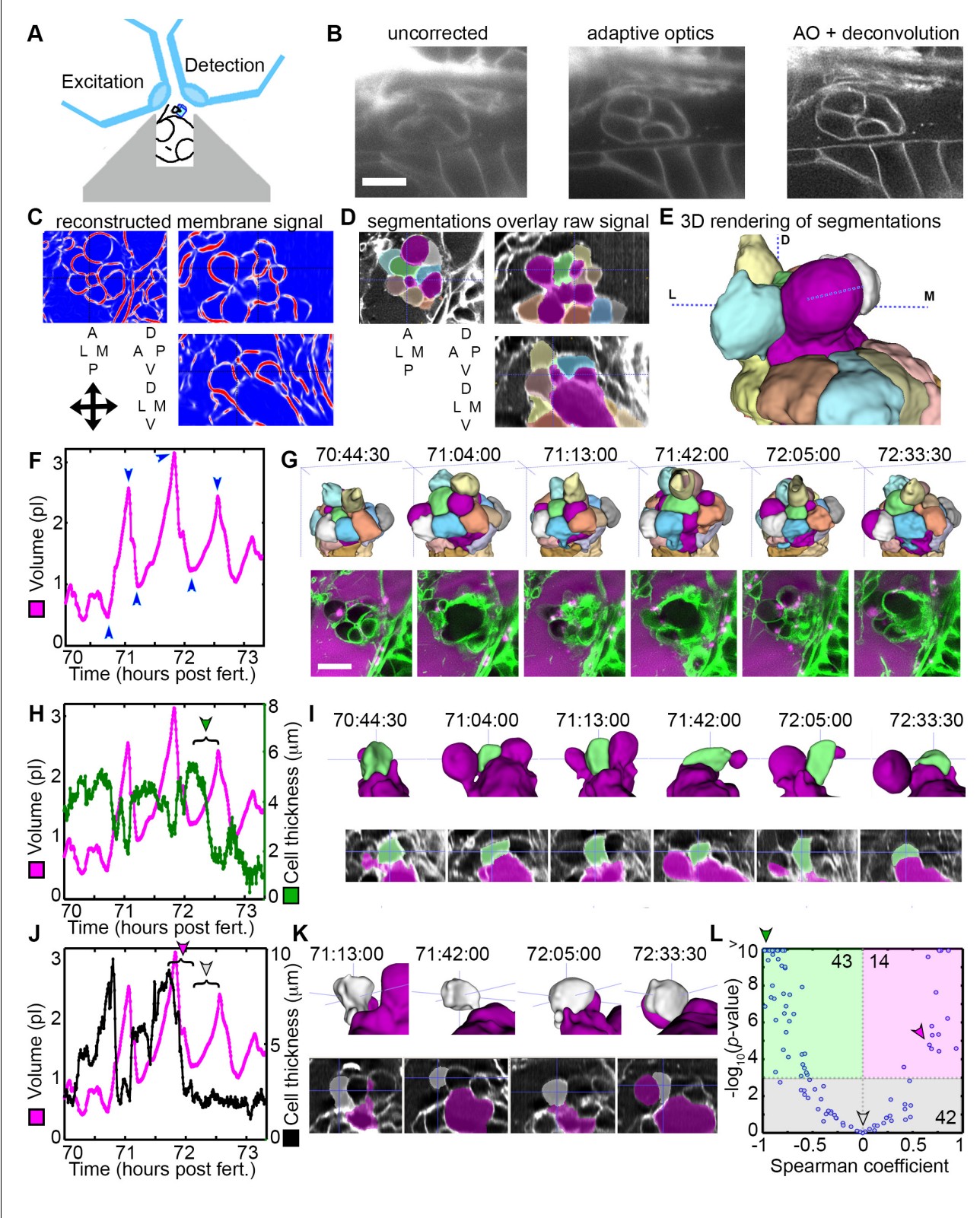

**Figure 5.** AO-LLSM reveals dynamics of ES cells. (**A**) Illustration of AO-LLSM mounting strategy for imaging ES using volcano mount. (**B**) Representative LLSM images without adaptive optics (AO), with AO, and with AO followed by deconvolution. (**C**) Three orthogonal views of ACME membrane reconstruction. (**D**) Three orthogonal views of raw fluorescence signal overlaid with cell segmentations and ES lumen segmentation (magenta), *n* = 4. (**E**) 3D rendering of segmented cells and ES lumen (magenta). (**F**) Volume measurements of segmented ES lumen, imaged every 30 s for over 3 hr

*Figure 5 continued on next page*

*Figure 5 continued*

(*Videos 8–10*). Blue arrowheads point to time points presented in G and I. (**G**) Top, dorsal perspective displaying 3D renderings of segmented cells. Bottom, maximum intensity projections (MIP) of 4.5 μm slab through tip of ES shows raw data of the ES for the same time points. (**H**) Secondary axis presents green cell's thickness versus time (green cell in (**G**)). Again, primary axis is volume of ES lumen for comparison. (**I**) Top, 3D renderings of just green cell from (**G**) and magenta lumen highlight stretching of cell, dorsal-medial perspective. Bottom, centered cross-sectional view of raw data overlaid with green cell's and ES lumen's segmentations for the same time points. (**J**) Secondary axis is plot of grey cell's thickness (grey in (**G**)). (**K**) Top, 3D renderings of only grey cell and magenta lumen. Bottom, centered cross-section view of raw data overlaid with grey cell's and ES lumen's segmentations for same time points. (**L**) Scatter plot of results from Spearman correlation test of cell thickness trajectories and ES lumen volume trajectories for individual inflation and deflation intervals that are monotonic (example intervals bracketed in H and J). Green region highlights significant correlation (p-value<$10^{-3}$) between cells thinning during inflation or thickening during deflation. Green arrowhead points to test result for bracketed interval in H. Grey region highlights instances where there is no significant correlation between the trajectory of cell thickness and lumen volume. Grey arrowhead points to test result for bracketed interval in J (grey arrowhead). Magenta region highlights significant correlation between cells thinning during deflation or thickening during inflation. Magenta arrowhead points to test result for bracketed interval in J (magenta arrowhead). Y-axis was capped at 10 so that all values greater than or equal to 10 plot as 10, *n* = 99. All scale bars 10 μm.

DOI: https://doi.org/10.7554/eLife.37131.017

Cell bodies within the ES stretch in response to increasing pressure. While the increased resolution of AO-LLSM did not allow us to resolve overlapping basal lamellae (thickness ~40 nm), it did enable segmentation of individual cell bodies (lengths and thicknesses ~1–10 μm). We found that some cells stretched and thinned when the ES lumen was inflated and thickened upon deflation, likely a result of their elastic properties (*Figure 5H,I*). However, there were instances when the cells at the tip of the ES did not thin during inflation, or while they thinned during some events, they did not thin during others (*Figure 5J,K*). To quantify these correlations and obtain an overview of the range of behaviors, we determined the Spearman correlation coefficient between trajectories of lumen volume and cell thickness for intervals that spanned individual inflation and deflation events (*Figure 5L*, bracketed examples in *Figure 5H,J*). For 43 of 99 tested trajectories, either cell thinning significantly correlated (*p*-value less than $10^{-3}$) with inflation or cell thickening significantly correlated with deflation (green region, *Figure 5L*, example of significantly correlated interval, bracket with green arrowhead, *Figure 5H*, same data point indicated with green arrowhead in *Figure 5L*). For 42 of 99 tested trajectories, cell behavior did not significantly correlate with inflation or deflation (grey region, *Figure 5L*, example of uncorrelated inflation interval, bracket with grey arrowhead, *Figure 5J*, same data point indicated with grey arrowhead in *Figure 5L*). Unexpectedly, for 14 of 99 tested trajectories there was significant correlation between inflation and cell thickening or deflation and cell thinning (magenta region, *Figure 5L*, example of correlated interval, bracket with magenta arrowhead, *Figure 5J*, same data point indicated with magenta arrowhead in *Figure 5L*). While each cell's behavior is varied, these data suggest that stretching of a subset of cells contributes to regulation of increased pressure by allowing the ES lumen to expand. Additionally, some cells appear to be pushed away from the ES lumen during inflation and pulled back into the ES epithelium during deflation. The complexity of the response is likely the result of features we did not resolve such as the dynamics of each cell's basal lamellae, each cell's residual apical and lateral adhesion,

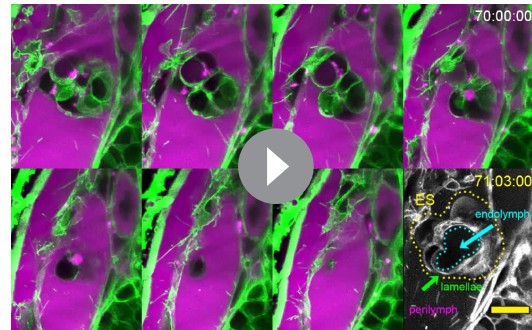

**Video 9.** Slab view of ES time course acquired with lattice light-sheet microscopy with adaptive optics. 15 sequential slices (300 nm slice spacing) were combined as a maximum intensity projections (MIP) to make a 4.5 μm slab. 7 sequential 4.5 μm slabs were tiled to consolidate the presentation of a complete 3D time course. The membrane citrine signal is green and the 3 kDa dextran-Texas red perilymph highlighter is magenta. Lower right panel is an annotated reference, with the basal surface of the ES labeled and outlined with a dotted yellow line, the apical interface enclosing endolymph outlined with a dotted blue line, the endolymph within the ES lumen indicated with a blue arrow, exposed basal lamellae indicated with a green arrow, and the perilymph labeled with magenta text. Scale bar is 10 μm.

DOI: https://doi.org/10.7554/eLife.37131.018

**Video 10.** 3D rendering of tracked and segmented cells and ES lumen. An anterior view on the left and dorsal view on the right. Segmented ES lumen is colored magenta. All other objects are ES cells. Labeled cubes indicate body axes. Same time course as *Video 9*.
DOI: https://doi.org/10.7554/eLife.37131.019

**Video 11.** 3D rendering of membrane citrine signal. The video begins with an annotated time point from the 3D rendering of signal from an AO-LLSM time course. A yellow dotted line highlights the rendered ES, green arrows point to basal lamellae, and the surrounding space is labeled as perilymph with magenta text. Dorsal view of ES, membrane citrine signal rendered in 3D. Scale bar is 10 µm.
DOI: https://doi.org/10.7554/eLife.37131.021

and each cell's basal interface with the extracellular matrix.

## Basal lamellae are dynamic

The low temporal resolution of our original confocal time courses made it unclear whether basal lamellae are static structures like floodgates, only opening to release endolymph, or dynamic. 3D rendering of the AO-LLSM signal shows that some lamellae are rapidly extending and retracting like they are crawling over neighboring cells or neighboring lamellae (*Video 11*). The basal lamellae are composed of two juxtaposed membranes and can overlay abutting membranes of adjacent cells or additional lamellae (*Figure 4*). While we observed stacked membranes by EM, they are still unresolved by AO-LLSM. However, one might expect patches of increased membrane signal several times the brightness of that of a lone cell membrane. In 3D renderings of the time course we see dynamic patches of increased intensity that can shift rapidly (*Figure 6A*, *Video 11*). By merging 3D renderings of 3 consecutive time points, 30 s apart, in red, blue, and then green, we observed the relative displacement of these lamellae that can crawl a few micrometers per minute (*Figure 6B,C*).

While the resolution of AO-LLSM is improved, it does not resolve the double membrane structure of lamellae or enable identification of the cell from which a lamella extends. Thus, in our surface-rendered segmentations, we give the cell membranes of each cell body a unique pseudo-color while the lamellae that are just exposed to the lumen and not adjacent to another cell body are collectively colored magenta (*Figure 5E,G*, corresponds to black dotted outline in *Figure 4F*). By visualizing the segmented objects in 3D we can identify when lamellar barriers are exposed based on when and where the magenta appears (*Figures 5G* and *6D*, *Video 10*). Inflations of the ES lumen coincided with slow increases in the surface area of exposed lamellae (30 min expansion, *Figure 6D*). The expansion of thin membrane signal was also observable in the raw data (bottom portion of panel, *Figure 6D*). Some secondary loci included smaller lamellae that engaged and inflated much more quickly (1–2 min expansion, *Figure 6E*). We can visualize their sudden expansion by again merging consecutive time points, 30 s apart, with red, blue, and then green (*Figure 6F*). This presentation reveals that the same locus can rapidly expand multiple times (one minute rapid expansions an hour apart, *Figure 6F,G*). In summary, multiple sets of basal lamellae can simultaneously engage at different loci, either through slow spreading or rapid expansion.

## Apical and basal myosin during ES inflation cycles

Cells and tissues modulate tension for crawling, retracting, bending, constricting, rounding-up, and resisting stretch by organizing an extensive variety of complexes of actin filaments, actin binding proteins, and myosin motors (*Grill, 2011*; *Munjal and Lecuit, 2014*; *Rafelski and Theriot, 2004*; *Stewart et al., 2011*). Myosin distribution correlates strongly with contractile forces (*Fournier et al., 2010*). Therefore, to begin to determine how myosin and actin might contribute to tension during ES inflation and deflation, we imaged myosin and actin using non-muscle myosin light chain fused to

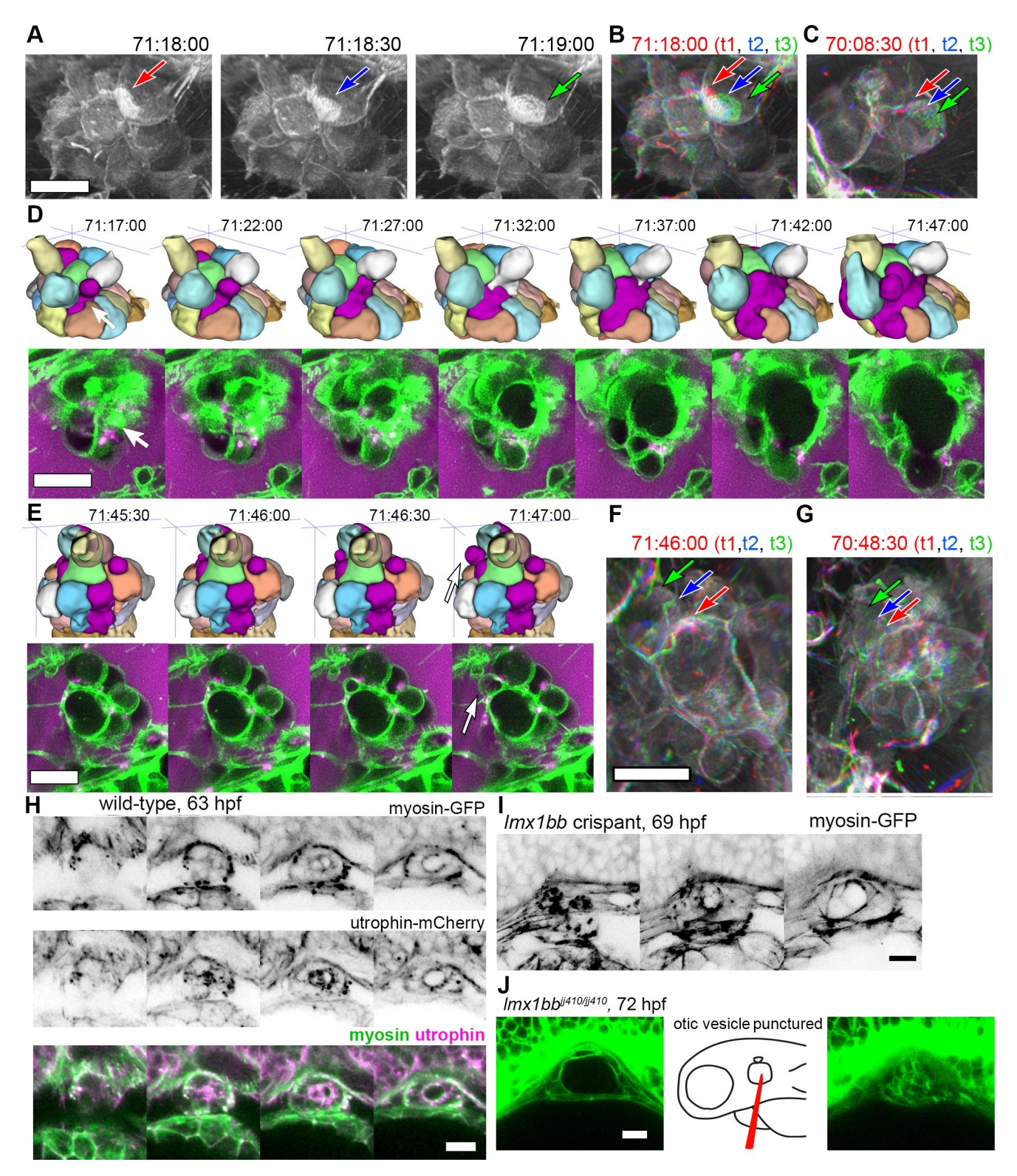

**Figure 6.** Basal lamellae are dynamic. (**A**) 3D rendering of AO-LLSM data, three sequential time points 30 s apart (membrane citrine depicted in grey, *Video 10*). Bright patches on surface move (red, blue, green arrows). (**B**) Consecutive 3D renderings overlaid as red (71:18:00), blue (71:18:30), and then green (71:19:00). Immobile regions remain grey, while regions of displacement are red, blue, and green. Arrows point to moving lamellae. (**C**) Additional example of moving lamellae, again visualized by overlaying consecutive images in red (70:08:30), blue (70:09:00), and then green (70:09:30). *Figure 6 continued on next page*

*Figure 6 continued*

Arrows point to moving lamellae. (D) Top, time points of 3D rendered segmentations spanning 30 min. Below, 4.5 μm MIP slabs of the raw data for the same time points. For both views, arrow points to same region where lumen segmentation is slowly exposed as thin lamellar region expands. (E) Top, time points of 3D rendered segmentations spanning 2 min. Below, 4.5 μm MIP slabs of the raw data for the same time points. For both views, arrow points to same region where lumen segmentation rapidly inflates as thin lamellar region expands. (F) Consecutive 3D renderings, same as (E), overlaid as red (71:46:00), blue (71:46:30), and then green (71:47:00). Arrows point to rapidly inflating lamellae. (G) Consecutive 3D renderings, overlaid as red (70:48:30), blue (70:49:00), and then green (71:49:30). Arrows point to rapidly inflating lamellae, same region as (F). (H) 4.2 μm MIP slabs of a time point from a time-lapse of wild-type embryos expressing myosin GFP (top) mCherry-utrophin (middle), and merged (myosin, green, utrophin, magenta, see *Video 12*), n = 4. (I) 4.2 μm MIP slabs of a time point from a time-lapse of *lmx1bb* crispant embryos expressing myosin GFP (see *Video 12*), n = 2. (J) Dilated mutant ES collapses following puncture of the otic vesicle with a tungsten needle, n = 5. All scale bars 10 μm.

DOI: https://doi.org/10.7554/eLife.37131.020

eGFP and utrophin, an actin binding protein, fused to mCherry (confocal microscopy, *Figure 6H*, *Video 12*). Myosin localized to the apical domains of ES cells as well as to dynamic puncta at the basal membrane throughout both inflation and deflation of the ES. During ES inflation, contraction of apical myosin likely counteracts the stress of hydrostatic pressure and maintains the integrity of apical junctions (*Rauzi et al., 2010*). Tension from apical myosin, strain from hydrostatic pressure, and the adhesion strength between junctional protein complexes likely determines the elastic limit of apical junctions. The crossing of this elastic limit may cause the small apical openings observed in *Figure 4G*, which ultimately communicates the otic vesicle's hydrostatic pressure to the lamellar protrusions. The dynamic spots of basal myosin could indicate two possible activities: contraction to retract lamellar protrusions as observed during cell migration or contraction at focal adhesions. Utrophin localized strongest at apical domains of ES cells (*Figure 6H*), but unfortunately bleached too quickly to generate any meaningful insights into the potential of actin localization dynamics. It had been shown that utrophin based actin highlighters do not localize to actin within lamellae (*Belin et al., 2014*), which explains the relatively low amounts of basal utrophin signal. In contrast, fluorescent phalloidin labels actin filament enrichment at both apical and basal domains of ES cells (not shown). Our observations of consistent apical localization of myosin throughout ES cycling suggests that the same contractility both counteracts hydrostatic pressure during inflation and drives deflation when hydrostatic pressure drops as the relief valve is opened. We did not observe any sudden changes in myosin localization or behavior that would indicate regulation for either inflation or deflation. Analysis of myosin localization in *lmx1bb* crispants (CRISPR-Cas9 knockout embryos) revealed similar apical localization and dynamic basal puncta (*Figure 6I*, *Video 12*, representative of 2 time-lapses). To determine whether the mutant ES tissue has elastic material properties, we punctured the otic vesicle with a tungsten needle and found that the distended ES rapidly collapsed (5/5 mutant puncture experiments caused ES collapse, *Figure 6K*). Global inhibition of actin or myosin with cytochalasin D or (S)-nitro-blebbistatin correlated with ES deflation. However, it is currently impossible to determine whether this is owed to hydrostatic pressure not being maintained by the otic vesicle epithelium.

Given the finding that ES lamellae are dynamic with myosin puncta and phalloidin stain, we asked whether Rac1, known to promote actin polymerization in lamellipodial protrusions, was important for ES valve function (*Waterman-Storer et al., 1999*). Heat-shock induction, throughout the embryo, of a dominant negative Rac1 at 56 hpf resulted in otic vesicles that became leaky and subsequently collapsed between 60 and 70 hpf (*Video 13*, 19/34 leaky and collapsed ears versus 1/12 in a heat-shock gfp control)(*Kardash et al., 2010*). In contrast,

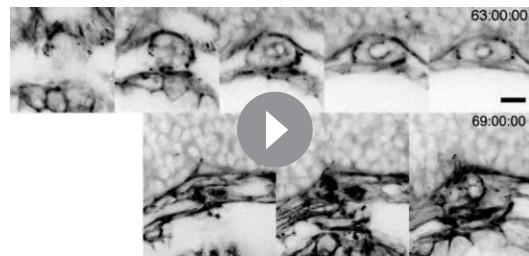

**Video 12.** Slab view of myosin-GFP during ES time course. Seven sequential slices (600 nm slice spacing) were combined as a maximum intensity projections (MIP) to make a 4.2 μm slabs. 4 sequential 4.2 μm slabs, starting with the distal tip of the ES on the left, were tiled to consolidate the presentation of a complete 3D time course. On top is the time lapse of a wild-type embryo expression myosin-GFP. Below is an *lmx1bb* crispant expressing myosin-GFP. Scale bar is 10 μm.

DOI: https://doi.org/10.7554/eLife.37131.022

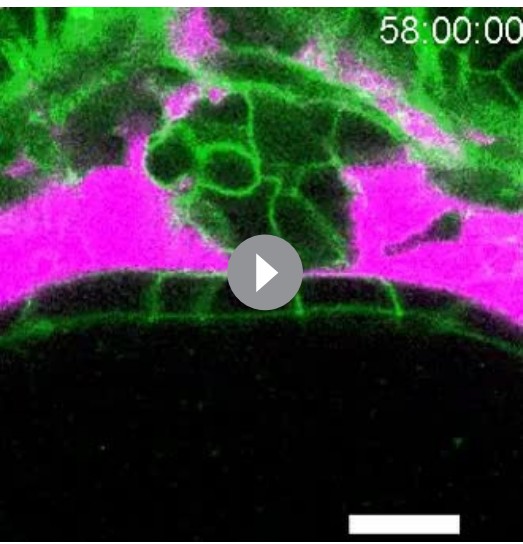

**Video 13.** Representative heat shock dnRac1 time course. Embryos were heat shocked at 55 hpf. At 57 hpf, α-bungarotoxin protein and 3 kDa dextran-Texas red were injected into the hearts. Time course began at 58 hpf, membrane citrine is green, perilymph is magenta. Scale bar is 10 μm.
DOI: https://doi.org/10.7554/eLife.37131.023

heat shock induction of the dominant negative Rac1 at 32 hpf, prior to ES formation, did not result in leaking or ear collapse during the subsequent 15 hr (while 0/20 leaky or collapsed ears versus and 0/14 in a heat-shock gfp control). These results are consistent with an important role for the basal lamellipodia of the ES for maintaining barrier integrity during inflation and deflation cycles. Precise analysis of the roles of actin and myosin networks during ES inflation cycles will require either the identification of regulators specific to the ES or actomyosin perturbations specific to the ES.

## Basal lamellae open for deflation

While we had evidence that lamellae can separate in our EM images and evidence that the epithelial barrier breaks during deflation, as seen in the leak in fluorescence from labeled perilymph, it remained uncertain whether separation of lamellae precipitated the deflation events. To observe evidence of the lamellar valve opening, we looked for instances when the membrane signal of lamellar barriers was disrupted prior to deflation events using AO-LLSM. We were able to witness instances when membrane signal from inflated lamellae was disrupted prior to deflation (yellow arrows, *Figure 7A–D*, lumen magenta, adjacent cells colored in at first at last time-points). These events can be accompanied by a thin lamellar protrusion sticking out into the perilymph (arrow, *Figure 7D*). Examination of the raw and rendered segmentations prior to and following breaks in membrane continuity showed that it preceded a deflation event (reduction in local volume of ES lumen, magenta, *Figure 7A,C*). To confirm that the small openings cause the rapid expulsion of endolymph during deflation events, we more closely examined time-lapse movies with labeled endolymph (*Figure 7E*, yellow endolymph volume before and after two rapid openings). Close examination of the path of endolymph release revealed a small opening through which endolymph travels out into the periotic space (x-y and y-z planes at relief valve openings, yellow arrows, *Figure 7E*). Narrow paths of pressure release were observed in all high time-resolution (20 s) time-lapse movies of endolymph dynamics (four fish in *Video 14* exhibit 16 opening events).

## Discussion

We report identification of the ES as a pressure relief valve based on six lines of evidence. First, the lumen of the ES slowly inflates and then rapidly deflates every ~0.3–4.4 hr. Second, deflation of the ES coincides with a breach in its epithelial diffusion barrier. Third, laser ablation of the otic epithelium is sufficient to induce rapid deflation of the inflating ES. Fourth, the ultrastructure of the closed and opened relief valves in the ES shows that lamellar barriers can pull apart. Fifth, high temporal and spatial resolution time course imaging reveal lamellae that behave like lamellipodia: constantly crawling over one another before they separate to relieve pressure and excess volume. Sixth, perturbation of the valve's development with a genetic mutation causes distension of the ES as found in common ear disorders such as Ménière's disease, Pendred syndrome, and Enlarged Vestibular Aqueduct syndrome (*Levenson et al., 1989*; *Jackler and De La Cruz, 1989*; *Belal and Antunez, 1980*; *Schuknecht and Gulya, 1983*).

Upon first observing the inflation and deflation cycles of the ES, we postulated that the underlying mechanism could either be a response to organ-wide hydrostatic pressure within the otic vesicle, a local tissue behavior where the ES inflates with fluid from the perilymph, or a local tissue behavior where cells in the ES periodically coordinate their movements to expand the ES volume. Our

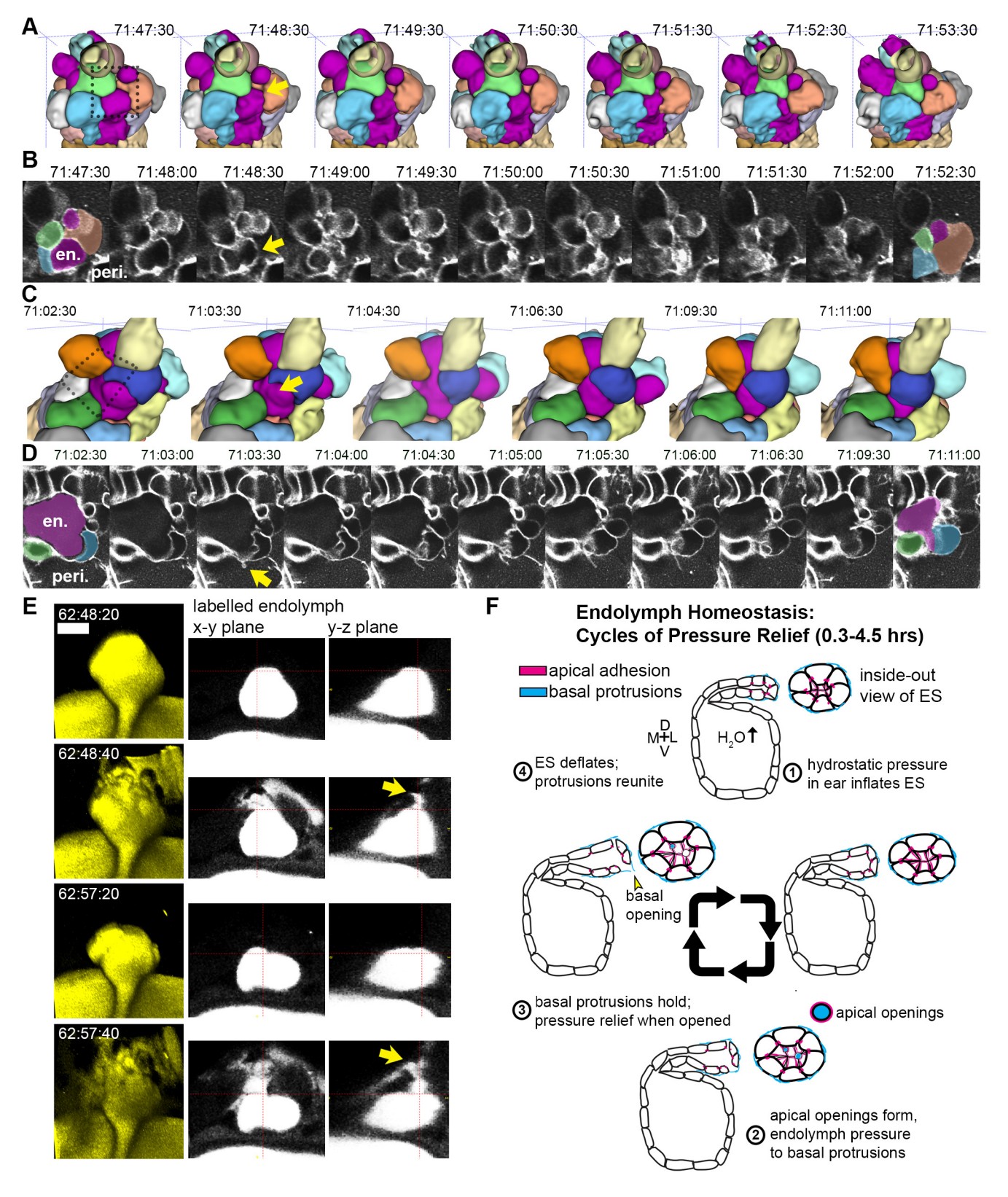

**Figure 7.** Basal lamellae open prior to deflation. (A,C) 3D renderings of segmented cells and ES lumen. (A) Dorsal view. (B) Raw data of membrane citrine AO-LLSM data spanning same time range as in (A). First and last time point are overlaid with segmentations for cells and lumen (magenta)

*Figure 7 continued on next page*

*Figure 7 continued*
neighboring the region of valve opening. In (**A–D**) arrows point to site of lamellae separating, dotted outlines indicate field of views in B and D, en. indicates endolymph lumen, peri. indicates perilymph. (**C**) Anterior view of another instance of lamellae separating. (**D**) Raw data spanning time range of (**C**). (**E**) Dorsal view of endolymph volume within ES lumen and time points before and after two release events. Middle and right panels show x-y and y-z ortho-planes of the isolated release sites, highlighted with yellow arrows (See *Video 14*), n = 8 (**F**) Illustrated pressure relief mechanism.
DOI: https://doi.org/10.7554/eLife.37131.024

ablation experiments indicated that pressure is transmitted to the ES from the OV and that the tissue is elastic because it collapses quickly when the stress from hydrostatic pressure is removed (*Figure 2E–F*). The second and third models lack support: dye tracing shows that during inflation the ES is filled with endolymph from the otic vesicle, while perilymph only enters the ES during deflation when the epithelium is open. We did not observe coordinated cell movements within the ES other than stretching. Additionally, myosin appears to act consistently throughout the ES cycles likely providing a constant apical tension that both resists pressure-induced stretch during inflation and drives collapse during deflation by contributing to tissue elasticity. This constant activity precludes the need for sensors or signals to tell the valve when to inflate, when to open, and when to collapse. Furthermore, dynamic basal puncta of myosin may provide integrity to adhesion between basal lamellae with added tension (*Ingber, 1997*). Throughout our imaging experiments, whether the contrast was produced at plasma membranes, endolymph, or perilymph, we observed what appear to be small pockets of lumen that pinch off from the rest of the ES lumen (see *Figures 2D,G*, *5G*, *6D–E* and *7E*, and *Videos 4*, *5*, *9*, *10*, *11* and *14*). These pockets were not present in the *lmx1bb* mutant, (*Figure 3E*). The simplest explanation is that fluid gets trapped in the lateral intracellular space as the apical junctions open but before the basal lamellae open or when the basal lamellae close after a deflation event. This is consistent with how the inclusions are often resolved in the next inflation cycle. Incorporating all of our observations, we surmise that increased pressure is managed through a combination of strain that stretches viscoelastic cells in the ES and adhesion distributed across the surface area of dynamic basal lamellae. Excess pressure and volume is then released when small openings form amongst compliant apical junctions of ES cells that transmits the stress of the ear's hydrostatic pressure to the basal lamellae that hold until they separate to release excess pressure and volume. The elastic properties of the tissue then drive its deflation, basal lamellae reunite, and the homeostatic cycle begins again (*Figure 7F*).

A physical relief valve in the ES, composed of overlapping basal lamellae, had not been previously identified because of the lack of sufficient temporal and spatial resolution as well as the lack of a system with an optically accessible ES. Thin overlapping basal junctions are seen at the marginal folds of lymph and blood capillaries (*Baluk et al., 2007*). Capillaries flank the developing and adult ES and the similarity between the cross-sectional ultrastructure of lamellar barriers and capillaries, as well as vacuoles, could have led to inaccurate annotation of lamellar barriers (*Kronenberg and Leventon, 1986*). EM

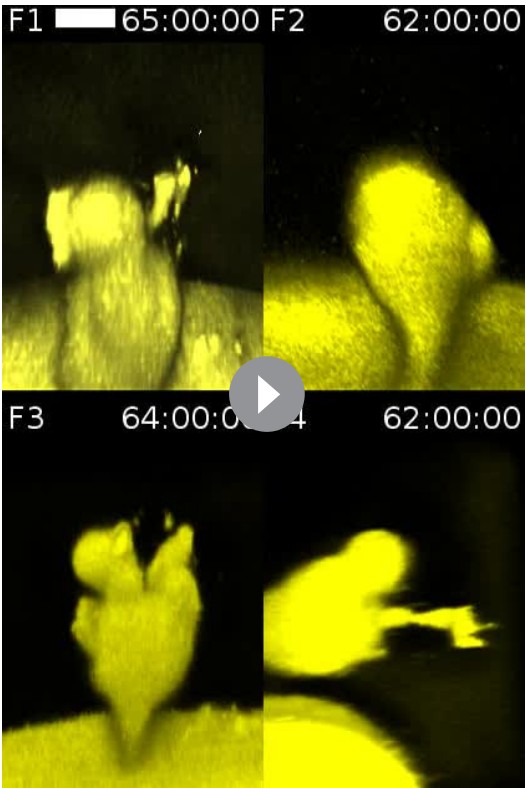

**Video 14.** Endolymph time course at high time resolution reveal sites of release. Four embryos with 3 kDa dextran-Texas red injected into the otic vesicle. First three are 3-D rendered volumes from a dorsal view. The fourth fish is 3-D rendered from a transverse view (as in *Videos 4* and *5*). Scale bar is 10 μm.
DOI: https://doi.org/10.7554/eLife.37131.025

micrographs of the rat, guinea pig, tree frog, and human ES showed instances of thin cytoplasmic extensions enclosing large lumens, which were annotated as capillaries or large vacuoles (*Bagger-SjobackSjöbäck et al., 1986*; *Bagger-Sjöbäck and Rask-Andersen, 1986*; *Dahlmann and von Düring, 1995*; *Kawamata et al., 1987*; *Møller et al., 2013*; *Qvortrup et al., 1999*). More recent EM studies of the human ES, which used more refined techniques to prepare samples from adult cadavers, recognized similar structures as belonging to the distal portion of the ES and described them as interconnected lumen resembling a network of cisterns (*Møller et al., 2013*). While appearing more elaborate in humans, the structural unit of the relief valve seems to be conserved. However, because of insufficient spatial and temporal resolution, it remains to be determined whether these cells have overlapping lamellae and whether they function as a physical relief valve.

Primarily, pressure relief prevents endolymphatic hydrops—the buildup of pressure and potential tearing of the ear's epithelium—that is associated with many inner ear pathologies. Secondarily, low-frequency deflations of the ES may drive intermittent longitudinal flow of endolymph from the cochlea and semicircular canals into the endolymphatic duct and sac. The presence of longitudinal flow had been proposed to explain the accumulation of debris and tracers within the ES and, while never directly observed in an unperturbed inner ear, longitudinal flow can be induced and observed through manipulations of the inner ear (*Salt, 2001*; *Salt and DeMott, 1998*; *Salt and Plontke, 2010*; *Salt et al., 1986*). The localized inflation and patterned formation of valve cells leads to a localized break at the ES lamellar barriers. The local break could create a transient pressure gradient that leads to flow from other parts of the inner ear towards the endolymphatic duct and sac. The intermittent and slow nature of this flow could explain conflicting interpretations of classic studies using tracer injections where tracer accumulated in the ES after long periods of time but significant flow was absent when observed on short time scales (*Guild, 1927*; *Manni and Kuijpers, 1987*; *Salt, 2001*; *Salt et al., 1986*). We observed longitudinal flow in action with the expulsion of leaked-in perilymph (*Figure 2G*, *Video 5*)

Lmx1bb is a LIM homeobox transcription factor. Mutations in *LMX1B* cause Nail Patella syndrome in humans and a third of Nail Patella syndrome patients develop glaucoma, a phenotype also observed in mouse models (*Cross et al., 2014*; *Liu and Johnson, 2010*; *Mimiwati et al., 2006*). The mechanism by which glaucoma arises in the disease and in mouse models is not clear; however, the trabecular meshwork and Schlemm's canal appear abnormal in these cases. Giant vacuoles and pores in Schlemm's canal may be used to relieve intraocular pressure, and their ultrastructure resembles the lamellar barriers of the ES (*Gong et al., 2002*). Like the ear, pulsatile pressure relief occurs in the eye, although at the higher frequency of the heart rate, and Schlemm's canal is a major site of the pressure relief (*Ascher, 1962*; *Johnstone et al., 2011*). Additionally, up to half of Nail Patella syndrome patients suffer from kidney disease (*Bennett et al., 1973*). During podocyte maturation apical junctions are lost, thereby enabling development of the basal slit diaphragm that filters blood (*Pavenstädt et al., 2003*). In mice lacking LMX1B activity, podocytes fail to lose their apical junctions (*Miner et al., 2002*). Removal of apical junctions may thus be a specialized mechanism used in organs where elaborations of junction architectures allow controlled release of fluids from one compartment into another. Furthermore, the mouse *Lmx1a* mutant exhibits defects (*Koo et al., 2009*) like the fish *lmx1bb* mutant in that they exhibit similarly enlarged ears. Similarity in its protein sequence and expression suggests that mouse *Lmx1a* and fish *lmx1bb* likely have comparable activities. We speculate that LMX1 transcription factors may have a common set of targets in the ES, eye, and kidney involved in apical junction remodeling to regulate fluid flow across epithelia.

We find that apical junctions and basal lamellae in the endolymphatic sac behave like a pressure relief valve to regulate fluctuating volume and pressure that arises from excess endolymph in the inner ear (*Figure 7F*). The high temporal and spatial resolution of AO-LLSM, combined with image processing tools for segmenting, tracking, and quantifying cell geometries, was necessary to reveal the dynamic cell behaviors underlying the pulsing of the ES. While cells of the ES are immobile epithelial cells, the cell extensions resemble lamellipodia of crawling cells in thickness (40 nm), speed (as fast as ~1 µm per minute), and regulatory mechanism (Rac1) (*Abercrombie, 1980*). Because of limitations of imaging, it remains unclear how each cell's basal lamellae, residual apical and lateral adhesion, and basal interface with the extracellular matrix contribute to the physical behavior of the ES. Of related interest is how the crawling lamellae adhere to their substrate in the ES to generate a barrier without gaps during the inflation phase. The force of this adhesion, in combination with the viscoelastic properties of the ES, likely determines the relief valve's set point for volume and

pressure homeostasis. For the inner ears of adult mammals, this set point is likely around 100–400 Pascals, although it is unknown how much the inner ear's pressure and volume fluctuate (*Park et al., 2012*). In our mutant over-inflation videos we observed an instance of an incomplete deflation (*Video 6*) that contrasts with wild-type deflation cycles and mutant ES collapse after the otic vesicle was punctured (*Figure 6J*). This difference raises the questions of how pathological tearing of otic epithelium from endolymphatic hydrops might lead to hearing and balance symptoms and whether the organization of the endolymphatic duct and sac prevents an influx of low potassium perilymph from coming into direct contact with the stereocilia of sensory hair cells. Determination of the molecular basis of lamellar barrier organization and apical junction dynamics will be necessary to make progress on how the physical relief valve works and how it might fail to cause disease.

# Materials and methods

## Key resources table

| Reagent type (species) or resource | Designation | Source or reference | Identifiers | Additional information |
|---|---|---|---|---|
| Strain, strain background (*Danio rerio*) | AB | ZIRC, Eugene, OR | ZFIN ID: ZDB-GENO-960809–7 | |
| Strain, strain background (*Danio rerio*) | *lmx1bb* mutant, ale uchu (*jj410* allele) | ZIRC, Eugene, OR, PMID: 17574823 | *jj410*; ZFIN ID: ZDB-ALT-070426–3 | *Schibler and Malicki, 2007* |
| Strain, strain background (*Danio rerio*) | *Tg(actb2:mem-citrine-citrine)$^{hm30}$* | Megason lab, PMID: 25303534 | *hm30*; ZFIN ID: ZDB-ALT-150209–1 | *Xiong et al., 2014* |
| Strain, strain background (*Danio rerio*) | *Tg(actb2:mem-citrine)/(actb2:Hsa.H2b-tdTomato)$^{hm32}$* | Megason lab, PMID: 27535432 | *hm32*; ZFIN ID: ZDB-ALT-161213–1 | *Aguet et al., 2016* |
| Strain, strain background (*Danio rerio*) | *Tg(actb2:mem-citrine)/(actb2:Hsa.H2b-tdTomato)$^{hm33}$* | Megason lab, PMID: 27535432 | *hm33*; ZFIN ID: ZDB-ALT-161213–2 | *Aguet et al., 2016* |
| Strain, strain background (*Danio rerio*) | *Tg(−5.0lmx1bb:d2eEGFP)$^{mw10}$* | gift from Brian Link's lab, PMID: 19500562 | *mw10*; ZFIN ID: ZDB-ALT-091218–2 | *McMahon et al., 2009* |
| Strain, strain background (*Danio rerio*) | *Tg(actb2:mem-mcherry2)$^{hm29}$* | Megason lab, PMID: 23622240 | *hm29*; ZFIN ID: ZDB-ALT-130625–1 | *Xiong et al., 2013* |
| Strain, strain background (*Danio rerio*) | *Tg(hsp70:rac1_T17N-p2a-mem-cherry2)$^{hm35}$* | Megason lab, *rac1* mutant plasmid gift from Raz lab, this paper | *hm35* | *Kardash et al., 2010* |
| Strain, strain background (*Danio rerio*) | *Tg(elavl3:GCaMP5G)$^{a4598}$* | gift from Alexander Schier's lab, PMID: 23524393 | *a4598*; ZFIN ID: ZDB-ALT-130924–1 | *Ahrens et al., 2013* |
| Strain, strain background (*Danio rerio*) | *Tg(actb2:myl12.1-EGFP)$^{e2212}$* | gift from C.P. Heisenberg's lab, PMID: 25535919 | *e2212*; ZFIN ID: ZDB-ALT-130108–2 | *Compagnon et al., 2014* |
| Strain, strain background (*Danio rerio*) | *Tg(actb2:mCherry-Hsa.UTRN)$^{e119}$* | gift from C.P. Heisenberg's lab, PMID: 25535919 | *e119*; ZFIN ID: ZDB-ALT-151029–2 | *Compagnon et al., 2014* |
| Antibody | mouse anti ZO-1 | Thermo Fisher Scientific, Waltham, MA | ZO1-1A12 | |
| Antibody | rabbit anti collagen II | Abcam, Cambridge, United Kingdom | ab209865 | |
| Antibody | rabbit anti laminin | Sigma-Aldrich, St. Louis, MO | L9393 | |
| Recombinant DNA reagent | pet-28b-Cas9-His | gift from Alexander Schier's lab, PMID: 24873830 | addgene id: 47327 | *Gagnon et al., 2014* |
| Recombinant DNA reagent | pmtb-t7-alpha-bungarotoxin | Megason lab, PMID: 26244658 | addgene id: 69542 | *Swinburne et al., 2015* |
| Sequence-based reagent | *foxi* in situ probes, *Danio rerio* | PCR template + T7 reaction (Sigma) | RefSeq:NM_181735 | *Thisse and Thisse, 2014* |

*Continued on next page*

*Continued*

| Reagent type (species) or resource | Designation | Source or reference | Identifiers | Additional information |
|---|---|---|---|---|
| Sequence-based reagent | *bmp4* in situ probes, *Danio rerio* | PCR template + T7 reaction (Sigma) | RefSeq:NM_131342 | *Thisse and Thisse, 2014* |
| Sequence-based reagent | *lmx1bb* sgRNA (exon 2), *Danio rerio* | annealed oligos + SP6 reaction (NEB) | GenBank:CR376762 | *Gagnon et al., 2014* |
| Sequence-based reagent | *lmx1bb* sgRNA (exon 3), *Danio rerio* | annealed oligos + SP6 reaction (NEB) | GenBank:CR376762 | *Gagnon et al., 2014* |
| Peptide, recombinant protein | cas9 protein | Megason lab | | purification scheme from *Gagnon et al., 2014* |
| Peptide, recombinant protein | alpha-bungarotoxin | Tocris Bioscience (Bristol, United Kingdom) | Tocris catalog number 2133 | *Swinburne et al., 2015* |
| Commercial assay or kit | mMessage mMachine T7 ULTRA kit | Thermo Fisher Scientific, Waltham, MA | AM1345 | |
| Chemical compound, drug | Dextran, Texas Red, 3000 MW | Thermo Fisher Scientific, Waltham, MA | D-3329 | |
| Chemical compound, drug | Dextran, Alexa Fluor 488, 10000 MW | Thermo Fisher Scientific, Waltham, MA | D-22913 | |
| Chemical compound, drug | tricaine methanosulfate | Sigma-Aldrich, St. Louis, MO | E10521 | |
| Chemical compound, drug | nonenyl succinic anhydride | Electron Microscopy Sciences, Hatfield, PA | 19050 | |
| Chemical compound, drug | DMP-30 | Electron Microscopy Sciences, Hatfield, PA | 13600 | |
| Chemical compound, drug | 1,2,7,8-diepoxyoctane (97%) | Sigma-Aldrich, St. Louis, MO | 139564 | |
| Chemical compound, drug | Sorensen's Phosphate Buffer | Electron Microscopy Sciences, Hatfield, PA | 11600–10 | |
| Chemical compound, drug | glutaraldehyde, EM grade | Electron Microscopy Sciences, Hatfield, PA | 16220 | |
| Chemical compound, drug | paraformaldehyde | Electron Microscopy Sciences, Hatfield, PA | 15710 | |
| Chemical compound, drug | potassium ferricyanide | Sigma-Aldrich, St. Louis, MO | 702587 | |
| Chemical compound, drug | osmium tetroxide | Electron Microscopy Sciences, Hatfield, PA | 19140 | |
| Chemical compound, drug | uranyl acetate | Electron Microscopy Sciences, Hatfield, PA | 22400 | |
| Chemical compound, drug | maleic acid | Sigma-Aldrich, St. Louis, MO | m5757 | |
| Chemical compound, drug | acetronitrile | Electron Microscopy Sciences, Hatfield, PA | 10020 | |
| Chemical compound, drug | Taab 812 Resin | Marivac Ltd., Nova Scotia, Canada | | |
| Software, algorithm | MovingROIExtract, convertFormat | https://github.com/krm15/AO-LLSM | | |
| Software, algorithm | ConvertToMegacapture, GoFigure2Contours ToMeshes | https://github.com/krm15/GF2Exchange | | |
| Software, algorithm | Itk-snap | www.itksnap.org PMID: 16545965 | | *Yushkevich et al., 2006* |
| Software, algorithm | Fluorender | www.sci.utah.edu/software/fluorender.html PMID:23584131 | | *Wan et al., 2012* |

*Continued on next page*

*Continued*

| Reagent type (species) or resource | Designation | Source or reference | Identifiers | Additional information |
|---|---|---|---|---|
| Software, algorithm | HandBrake | https://handbrake.fr/ | | |
| Software, algorithm | LabVIEW | National Instruments | | |
| Software, algorithm | MATLAB (R2014A) | www.mathworks.com | | |
| Software, algorithm | ParaView | www.paraview.org | | |
| Software, algorithm | GoFigure2 | | | *Xiong et al., 2013* |
| Software, algorithm | FIJI (imagJ) | www.fiji.sc PMID: 22743772 | | *Schindelin et al., 2012* |
| Software, algorithm | cellPreprocess, multiscale PlanarityAndVoting3D, DistanceFromMask, resample, MorphologicalErosion OnLabelImageFilter, SizeThreshold, Membrane Segmentation, Membrane SegmentationWithMarkers ImageFilter, CellSegmentation Statistics | https://github.com/krm15/ACME/tree/Multithread Lookup | | |
| Software, algorithm | Zen software | http://www.zeiss.com/microscopy/us/products/microscope-software/zen-lite.html | | |
| Other | Volcano mould, 'frosted extreme detail' | https://www.shapeways.com/, this paper | Volcano400 | |
| Other | FemtoJet 4x | Eppendorf, Hamburg, Germany | 5253000017 | |
| Other | Nanoject | Drummond Scientific, Broomall, PA | 3-000-204 | |
| Other | Tungsten wire | Sigma-Aldrich, St. Louis, MO | 267554–9.5G | |

## Zebrafish strains and maintenance

Zebrafish were maintained at 28.5°C using standard protocols (*Westerfield, 1993*). The Harvard Medical Area Standing Committee on Animals approved zebrafish work under protocol number 04487. Adult zebrafish, 3 months to 2 years of age, were mated to produce embryos and larvae. Adults were housed in a main fish facility containing a 19-rack Aquaneering system (San Diego, CA) and a 5-rack quarantine facility for strains entering the system from outside labs or stock centers. The systems' lights turn on at 9am and go off at 11pm. Fish matings were set up the night before with males separated from females with a divider in false-bottom crossing cages in a pair wise, or 2 × 2 fashion to maximize embryo yield. The divider was pulled the following morning, shortly after the lights turned on. Egg production was monitored to establish the time of fertilization. Manipulations and observations of baby zebrafish were performed between fertilization and 5.5 days post fertilization.

These studies were performed using the AB wild-type strain, the *lmx1bb*[*jj410/jj410*] mutant and the following transgenic lines: *Tg(actb2:mem-citrine-citrine)*[*hm30*], *Tg(actb2:mem-citrine)/(actb2:Hsa.H2b-tdTomato)*[*hm32*], *Tg(actb2:mem-citrine)/(actb2:Hsa.H2b-tdTomato)*[*hm33*] (these three alleles were combined for maximal membrane signal, *hm32* and *hm33* are two separate alleles of the same construct that is composed of two divergent beta-actin promoters, one driving membrane citrine and the other driving histone tdTomato, *actb2:Hsa.H2b-tdTomato* of these divergent constructs tends to be silenced in transgenic fish and is not useful), *Tg(−5.0lmx1bb:d2eEGFP)*[*mw10*], *Tg(actb2:mem-mcherry2)*[*hm29*], *Tg(hsp70:rac1_T17N-p2a-mem-cherry2)*[*hm35*], *Tg(elavl3:GCaMP5G)*[*a4598*], *Tg(actb2:myl12.1-EGFP)*[*e2212*], and *Tg(actb2:mCherry-Hsa.UTRN)*[*e119*] (*Aguet et al., 2016*; *Ahrens et al., 2013*;

Compagnon et al., 2014; McMahon et al., 2009; Obholzer et al., 2012; Schibler and Malicki, 2007; Xiong et al., 2014; Xiong et al., 2013).

## In situ and fluorescent immunostains

The *foxi1* in situs that confirm the identity of the ES tissue (*Figure 1B'*) is representative of 25 stained embryos. A similar ES tissue labeling was obtained when staining for *bmp4* (12 embryos at 55hpf). In situs were performed as previously described (*Thisse and Thisse, 2014*). The *foxi1* probe was made using T7 RNA polymerase (Sigma) off of a template made from cDNA using these PCR primers: ctccATGTTTCTGGAGGGAGAG and TAATACGACTCACTATAGGGagaGATCCGTCCCGGTTGTATATGAG. The *bmp4* probe was made using T7 RNA polymerase (Sigma) off of a template made from cDNA using these PCR primers: GTCGAGACATCATGATTCCTGG and TAATACGACTCACTATAGGG aga GAGTCTCCGTTTAGCGGCAGC.

Whole-mount fluorescent immunostains were performed using standard protocols and a mouse monoclonal antibody against ZO-1 (ZO1-1A12, Thermo Fisher Scientific, Waltham, MA), a rabbit polyclonal antibody against collagen II (ab209865, Abcam), and a rabbit polyclonal antibody against laminin (L9393, Sigma). In *Figure 3J*, the 50 hpf wt collagen stain is representative of 16 stained and imaged embryos, the 50 hpf wt ZO-1 stain is representative of 22 stained and imaged embryos, the 80 hpf wt collagen stain is representative of 11 stained and imaged embryos, the 80 hpf wt ZO-1 stain is representative of 32 stained and imaged embryos, the 80 hpf mutant collagen stain is representative of 7 stained and imaged embryos, and the 80 hpf mutant ZO-1 stain is representative of 6 stained and imaged embryos. In all stained embryos, the membrane contrast comes from the membrane citrine transgenic alleles.

Crispants for *lmx1bb* in the *Tg(actb2:myl12.1-EGFP)*[e2212] background were made using previously described approaches (*Gagnon et al., 2014*). sgRNA's were designed and synthesized to target exon 2 (GTTGCTTGTCCCGACCGCAGCGG) and exon 3 (ACGGAGTGCCATCACCAGGCGG) of *lmx1bb*. These guides were pooled and combined with purified Cas9 protein and injected into 1 cell stage embryos. Analyses were performed on F0 crispants that exhibited the full mutant phenotype.

## Immobilization and time-lapse imaging

Embryos were immobilized with either 50 pg of α-bungarotoxin mRNA injected into the 1 cell embryo or 2.8 ng of α-bungarotoxin protein injected into the heart at 59 hpf (*Swinburne et al., 2015*). α-bungarotoxin mRNA was synthesized from a linearized plasmid using the mMessage mMachine T7 ULTRA kit (Thermo Fisher Scientific). Subsequently, mRNA was purified using RNAeasy Mini Kit (Qiagen, Hilden, Germany). α-bungarotoxin protein was obtained from Tocris Bioscience (Bristol, United Kingdom). 2.3 nL injections were performed using Nanoject II (Drummond Scientific, Broomall, PA). For tracking perilymph,~10 ng of 3 kDa dextran-Texas red neutral (Thermo Fisher Scientific) was injected into the zebrafish heart at 59 hpf. For tracking endolymph,~1 ng of 3 kDa dextran-Texas red neutral was injected into the otic vesicle using a FemtoJet 4x (Eppendorf, Hamburg, Germany) to minimize the volume injected to ~0.2 nl. When labeling both endolymph and perilymph, 3 kDa dextran-Texas red neutral was used for endolymph and 10 kDa dextran- Alexa Fluor 488 (Thermo Fisher Scientific) was used for perilymph. Immobilization with α-bungarotoxin permits the ear to grow and develop normally (*Swinburne et al., 2015*).

Paralyzed embryos were cultured in Danieau buffer and mounted in an immersed 1.5% w/v agarose canyon mount or the volcano mount for AO-LLSM, that were generated from custom-made molds. The canyons were 0.4 mm wide and 1.5 mm deep. The volcano mold for AO-LLSM was printed by shapeways.com (New York City, NY). The inner width of the volcano was also 0.4 mm and was printed using their 'Frosted Extreme Detail' material. The embryo was placed dorso-laterally in the canyon or volcano so that dorsal portion of the left otic vesicle was either flush with a #1 coverslip placed over the canyon (embryos were tilted approximately 30 degrees laterally from their dorsal axes) or protruding from the mouth of the volcano mold. No coverslip was placed above embryos for AO-LLSM. Homemade hair loops were used to position embryos.

Mounted embryos were imaged on an upright LSM 710 (Carl ZEISS, Göttingen, Germany) with a C-apochromat 40x/NA 1.2 objective (ZEISS). The objective's corrective ring was adjusted to account for the use of #1 coverslips (0.13–0.16 mm thick). Imaging took place within a homemade foam-core incubator maintained at 28.5°C. Lasers of wavelength 405, 488, 514, and 594 nm were used to image

GFP, citrine, Texas red, and mCherry2. A typical imaging session used the following set up: citrine excited with 514 nm laser 5% (Ch1 filters 519–584 nm, gain 885), Texas red excited with 594 nm laser 15% (Ch2 filters 599–690 nm, gain 864), 1.27 µs pixel dwell, one line average, 137 µm pinhole, 1.2 zoom, 458/514/594 beam splitter, and 0.173 × 0.173×1.135 µm voxel scaling. Around 20–30 long-term time courses were attempted to establish the method. The data presented are representative of 8 wild-type time courses and four mutant time courses. Additionally, hundreds of ES distensions in mutants were observed during the cloning of the mutant and phenotype characterization at 72 and 96 hpf.

Laser ablations of epithelial cells in the otic vesicle were performed using a similar imaging set-up as for time-lapse confocal imaging. For the cell ablations a Mai-Tai HP 2-photon laser (Spectra-Physics, Santa Clara, CA) was used. After a target was chosen the 2-photon laser was tuned to 800 nm at 50% power, the pinhole was opened completely, the 690 + beam splitter was selected, and a spot scan was performed for 10,000 cycles. 2 or three spots were targeted in adjacent cells to ensure the wounds disrupted the epithelial barrier. After ablations, time-lapse microscopy was performed with 1-photon microscopy, as described above. The data presented are representative of 4 such experiments as well as seven ablations that were evaluated without time courses, 1 hr after ablation.

For puncturing with a tungsten needle, 78 hpf mutant ears were imaged on the confocal microscope, taken from the microscope to a dissecting microscope where the otic vesicle was rapidly punctured, remounted, and then reimaged on the confocal microscope within 2 min. 0.25 mm tungsten wire (Sigma) was sharpened by electrolysis to make tungsten needles.

For imaging with the adaptive optics lattice light-sheet microscope, zebrafish immobilized with α-bungarotoxin were mounted in a 1.5% low melt agarose volcano mold on a 5 mm coverslip and imaged starting at 68–72 hpf. We submerged the excitation and detection objectives along with the 5 mm coverslip in ~8 ml of 1X Danieau buffer at room temperature (22 ± 2°C). The zebrafish tissues expressing membrane citrine marker and 3 kDa dextran Texas Red fluid phase marker were excited using 488 nm and 560 nm lasers (488 nm operating at ~3 mW and 560 nm operating at 5 mW corresponding to ~16 µW and ~31 µW at the back aperture of the excitation objective) both sequentially exposed for 15 msec. The tissues were excited with 488 nm and 560 nm sequentially by dithering a multi-Bessel light-sheet arranged in a square lattice configuration (corresponding to an excitation inner/outer numerical aperture of 0.517/0.55, respectively). The optical sections were collected by scanning the detection objective with 300 nm steps. Each imaging volume consisted of 131 optical planes capturing 1024 × 1024 pixels, thereby capturing a volume of ~97 µm x 97 µm x 39 µm every 15.5 or 29.9 s using a dual Hamamatsu ORCA-Flash 4.0 sCMOS camera setup (Hamamatsu Photonics, Hamamatsu City, Japan). Prior to the acquisition of the time series data consisting of 300–530 time points, the imaged volume was corrected for optical aberrations (manuscript in preparation). The imaged volumes were deconvolved with experimentally measured point-spread functions measured with 0.17 µm tetraspec beads (Thermo Fisher) excited with 488 nm and 560 nm lasers in MATLAB using Lucy-Richardson algorithm on HHMI Janelia Research Campus' computing cluster and locally with a 3.3 GHz 32-Core Intel Xeon E5 with 512 GB memory. The LLSM was operated using a custom LabVIEW software (National Instruments, Woburn, MA) on a 3.47 GHz Intel Xeon X5690 workstation with 96 GB memory running Microsoft Windows seven operating system. The images presented are from one AO-LLSM time course that is representative of three other similar acquisitions also obtained using AO-LLSM.

## Serial-section electron microscopy

'Wild-type' *Tg(elavl3:GCaMP5G)*[a4598] and 'mutant' *lmx1bb*[jj410/jj410] larvae at 5.5 days post-fertilization (or 4.5 days for the mutant that begins to suffer from pleiotropic toxicity) were briefly anesthetized with 0.02% tricaine methanesulfonate (Sigma-Aldrich, St. Louis, MO), and subsequently preserved by dissection and immersion into an aldehyde fixative solution (2% paraformaldehyde and 2.5% glutaraldehyde in 0.08M Sorenson's phosphate buffer, Electron Microscopy Sciences, Hatfield, PA). Each specimen was then post-fixed and stained *en bloc* with reduced osmium solution (1% osmium tetroxide (Electron Microscopy Sciences) and 1.5% potassium ferricyanide (Sigma-Aldrich, St. Louis, MO)) followed by uranyl acetate (1% uranyl acetate (Electron Microscopy Sciences) in 0.05 M maleate buffer). Finally, samples were dehydrated with serial dilutions of acetonitrile in distilled water, infiltrated with serial dilutions of epoxy resin in acetonitrile (Electron Microscopy Sciences), and embedded in low-viscosity resin (*Koehler and Bullivant, 1973*). The epoxy resin for

infiltrations and embedding was composed of 63% nonenyl succinic anhydride (Electron Microscopy Sciences), 35.5% 1,2,7,8-diepoxyoctane (97%, Sigma), and 1.5% 2,4,6-tri(dimethylaminomethyl) phenol (DMP-30, Electron Microscopy Sciences), v/v.

Serial sections through the endolymphatic sac were continuously cut with a nominal thickness of 60 nm using a 45° diamond knife (Diatome, Biel, Switzerland) affixed to an ultramicrotome (Leica EM UC6, Leica Microsystems, Wetzlar, Germany) and collected with an automated tape-collecting ultramicrotome (*Hayworth et al., 2014*). Field-emission scanning EM of back-scattered electrons was conducted either on a Zeiss Merlin (ZEISS) or a FEI Magellan XHR 400L (Thermo Fisher Scientific) with an accelerating voltage of 5.0 kV and beam current of 1.6–7 nA (*Hildebrand et al., 2017*). Image registration was performed with Fiji TrakEM2 alignment plug-ins (*Saalfeld et al., 2010*; *Saalfeld et al., 2012*; *Schindelin et al., 2012*). Manual image segmentation was performed with ITK-SNAP (*Yushkevich et al., 2006*). The data presented are from a single wild-type and single mutant sample because of the time and effort required technology development as well as in sample preparation, acquisition, and processing.

Additional wild-type and *lmx1bb^{ij410/j4410}* mutant specimens were analyzed by transmission electron microscopy at the Harvard Medical School EM facility. Samples were fixed in 2.5% paraformaldehyde, 2.5% glutaraldehyde, 0.03% picric acid in 0.1M sodium cacodylate buffer, pH 7.4. Samples were stained for three hours in 1% osmium tetroxide and 1.5% potassium ferrocyanide, washed, and then stained with 1% uranyl acetate in maleate buffer, pH 5.2 for one hour. Samples were then embedded in Taab 812 Resin (Marivac Ltd., Nova Scotia, Canada). Blocks were sectioned with a Leica ultracut microtome to a thickness of 80 nm. Sections were collected on carbon coated, formvar coated, copper slot grids. Sections were further stained with with 0.2% lead citrate prior to viewing on a JEOL 1200EX (JEOL USA, Peabody, MA) using an AMT 2 k CCD camera (Advanced Microscopy Techniques, Corp., Woburn, MA).

## Quantification and statistical analyses

Time-lapse confocal data sets were converted from Zeiss's LSM format to a series of image files (*. png) with a header file containing information on the imaging set up (*.meg). This was performed with a script called *lsmtomegacapture* available in our open source 'in toto image analysis tool-kit' (ITIAT, https://wiki.med.harvard.edu/SysBio/Megason/GoFigureImageAnalysis). The image data sets were then loaded into GoFigure2, open-source software developed in the Megason lab for image analyses with a database (http://www.gofigure2.org; *Xiong et al., 2013*). In GoFigure2 the ES was manually contoured and these contours were exported as XML-files in the GFX format. Exported contours were modeled in 3D and volumes were quantified using a script called *GoFigure2ContoursToMeshes*, which uses the power crust reconstruction algorithm (*Amenta et al., 2001*). 3D meshes were generated in the VTK format and viewed using ParaView (KitWare) (*Henderson, 2007*). Leak in fluorescence was measured by placing a sphere with radius 1 µm into the lumen of the ES and GoFigure2 recorded the integrated fluorescence intensity.

Automated image analysis of the AO-LLSM data was performed using modified versions of the ACME software, python shell scripts, and ITIAT scripts (*Mosaliganti et al., 2012*). Starting with 3D tif files, the ACME pipeline includes the following steps, parameters used in (**bold**):

1. *convertFormat inputFile.tif outputFile.mha spacingX(**0.097**) spacingY(**0.097**) spacingZ(**0.3**)*
2. *cellPreprocess inputFile.mha outputFile.mha cellRadius (**0.1**)*
3. *resample inputFile.mha outputFile.mha spacingFactorX(**1.5**) spacingFactorY(**1.5**) spacingFactorY(**0.5**)*
4. *multiscalePlanarityAndVoting3D inputFile.mha outputFile.mha iSigmaFiltering(**0.25**) iSigmaVoting(**0.5**) alpha(**0.3**) beta(**0.25**) gamma(**1000**)*
5. *membraneSegmentation inputResample.mha inputTensorVoting.mha outputSegmentedLabelImage.mha threshold(**1.0**)*

Correcting of segmentations and tracking were done using iterative implementations of the ITIAT scripts *membraneSegmentationWithMarkersImageFilter* and *MorpholigcalErosionLabelImageFilter* with manual error corrections done in ITK-SNAP (*Yushkevich et al., 2006*).

3D rendering of segmented labels and overlays of segmented labels and raw membrane signal were performed using the software ITK-SNAP (www.itksnap.org, [*Yushkevich et al., 2006*]). 3D visualization of raw membrane data was performed using the software FluoRender (www.sci.utah.edu/

software/fluorender.html, [*Wan et al., 2012*]). Videos were compiled and encoded using a combination of ImageJ and HandBrake ([*Schindelin et al., 2012*], www.handbrake.fr).

For tracking cell thickness, we used the ITIAT scripts *CellSegmentationStatistics* to map the coordinates of each cell's centroid, *sizeThresh* to extract label image files of just the ES lumen, and *DistanceFromMask* to calculate the physical distance from cell centroids to the surface of the ES lumen labe, which was then multiplied by two to estimate the height of each cell. Cell and lumen volumes were also determined using *CellSegmentationStatistics.*

The outputs from *DistanceFromMask* and *CellSegmentationStatistics* were loaded into MATLAB for both graphing and further analysis. For determining the Spearman correlation coefficient between lumen volumes we first parsed the data matrix into time windows when lumen volume exhibited approximately monotonic increasing or decreasing values. For these time windows we used MATLAB's statistical toolbox to determine the Spearman correlation coefficient and its associate *p*-value. The syntax for this analysis was: [RHO,PVAL]=corr(LumenVolume,CellThickness,'type','Spearman'). This test was performed on 99 data windows. MATLAB was also used to perform a Mann-Whitney-Wilcoxon test on the significance cell thickness differences between wild-type and mutant images (*Figure 3*).

## Data and software availability

The image processing scripts described above are available through Github at the following repositories: https://github.com/krm15/ACME/tree/MultithreadLookup (*Mosaliganti, 2016*; copy archived at https://github.com/elifesciences-publications/ACME), https://github.com/krm15/AO-LLSM (*Mosaliganti, 2017a*; copy archived at https://github.com/elifesciences-publications/AO-LLSM) and https://github.com/krm15/GF2Exchange (*Mosaliganti, 2017b*; copy archived at https://github.com/elifesciences-publications/GF2Exchange)

The full data-set of serial-section EM images are publicly available at resolutions of $4.0 \times 4.0 \times 60$ $nm^3$ per voxel for the closed valve and $18.8 \times 18.8 \times 60$ $nm^3$ per voxel for the open valve. To examine the ES anatomy viewers should navigate to http://zebrafish.link/hildebrand16/data/es_closed (closed valve) or http://zebrafish.link/hildebrand16/data/es_open (open valve). This set of multi-resolution serial-section EM image volumes was co-registered (*Wetzel et al., 2016*) to allow one to navigate the lumen of the endolymphatic duct from the lumen of the otic vesicle to the tip of the ES.

## Acknowledgements

We thank Dante D'India for fish care. We thank Brian Link and Erez Raz for reagents. We thank Louise Trakimas, Elizabeth Benecchi, and Margaret Coughlin for assistance with electron microscopy at the Harvard Medical School EM facility. Lisa Goodrich, Becky Ward, Amelia Green, Akankshi Munjal, Sandra Swinburne, and the Megason lab for comments. Kerry Sobieski, Betzig lab members, and members of the Janelia Research Campus aquatics facility for helping with travel, shipping, and assisting with experiments at Janelia Research Campus. IAS was supported by NIH fellowship 5F32HL097599, a Hearing Health Foundation Emerging Research Grant, and a Novartis Fellowship in Systems Biology. KRM was supported by NIH grant K25 HD071969. DGCH was supported by NIH grants T32 MH20017 and T32 HL007901 and NSF IIA EAPSI award 1317014. TK acknowledges support from the Janelia Visitor Program, HHMI, Biogen and NIH grant R01 GM075252. SU is a Fellow at the Image and Data Analysis Core at Harvard Medical School. This work was supported by R01 DC010791 and R01 DC015478 from the National Institute of Deafness and Other Communication Disorders (SGM) and by NIH grants DP1 NS082121, RC2 NS069407 from the NINDS (FE).

## Additional information

### Funding

| Funder | Grant reference number | Author |
|---|---|---|
| National Institutes of Health | Fellowship 5F32HL097599 | Ian A Swinburne |
| Hearing Health Foundation | Emerging Research Grant | Ian A Swinburne |

| | | |
|---|---|---|
| Novartis | Fellowship in Systems Biology | Ian A Swinburne |
| National Institutes of Health | K25 HD071969 | Kishore R Mosaliganti |
| National Institutes of Health | T32 MH20017 | David G C Hildebrand |
| National Institutes of Health | T32 HL007901 | David G C Hildebrand |
| National Science Foundation | IIA EAPSI Award (1317014) | David G C Hildebrand |
| National Institutes of Health | DP1 NS082121 | Florian Engert |
| National Institutes of Health | RC2 NS069407 | Florian Engert |
| National Institute of Neurological Disorders and Stroke | | Florian Engert |
| National Institutes of Health | R01 GM075252 | Tomas Kirchausen |
| Janelia Visitor Program | | Tomas Kirchausen |
| Howard Hughes Medical Institute | | Tomas Kirchausen |
| Biogen | | Tomas Kirchausen |
| National Institutes of Health | R01 DC010791 | Sean G Megason |
| National Institutes of Health | R01 DC015478 | Sean G Megason |
| National Institute of Deafness and Other Communication Disorders | | Sean G Megason |

The funders had no role in study design, data collection and interpretation, or the decision to submit the work for publication.

## Author contributions

Ian A Swinburne, Conceptualization, Funding acquisition, Investigation, Writing—original draft, Writing—review and editing; Kishore R Mosaliganti, Srigokul Upadhyayula, Tony Y -C Tsai, Anzhi Chen, Ebaa Al-Obeidi, Anna K Fass, Samir Malhotra, Investigation; Tsung-Li Liu, Resources, Investigation; David G C Hildebrand, Florian Engert, Jeff W Lichtman, Resources; Tomas Kirchhausen, Resources, Supervision, Funding acquisition, Writing—review and editing; Eric Betzig, Resources, Supervision, Funding acquisition; Sean G Megason, Conceptualization, Supervision, Funding acquisition, Writing—original draft, Project administration, Writing—review and editing

## Author ORCIDs

Ian A Swinburne ID http://orcid.org/0000-0003-4162-0508
Srigokul Upadhyayula ID http://orcid.org/0000-0002-6911-0270
David G C Hildebrand ID https://orcid.org/0000-0002-2079-3517
Jeff W Lichtman ID http://orcid.org/0000-0002-0208-3212
Sean G Megason ID http://orcid.org/0000-0002-9330-2934

## Ethics

Animal experimentation: This study was performed in strict accordance with the recommendations in the Guide for the Care and Use of Laboratory Animals of the National Institutes of Health. The Harvard Medical Area Standing Committee on Animals approved zebrafish work under protocol number 04487

## Decision letter and Author response

Decision letter https://doi.org/10.7554/eLife.37131.030
Author response https://doi.org/10.7554/eLife.37131.031

# Additional files

## Supplementary files

• Transparent reporting form
DOI: https://doi.org/10.7554/eLife.37131.026

## Data availability

All data generated or analysed during this study are included in the manuscript and supporting files. The serial section electron microscopy data are available at http://zebrafish.link/hildebrand16 which is described fully in Hildebrand et al. (2017); ES-specific links presented in methods.

The following previously published dataset was used:

| Author(s) | Year | Dataset title | Dataset URL | Database, license, and accessibility information |
|---|---|---|---|---|
| Hildebrand DGC | 2017 | Multi-resolution serial-section electron microscopy data set containing the anterior quarter of a 5.5 days post fertilization larval zebrafish | http://zebrafish.link/hildebrand16 | Publicly available at NeuroData under an Open Data Commons Attribution License |

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
