## [Decision Letter]

[Editors’ note: a previous version of this study was rejected after peer review, but the authors submitted for reconsideration. The first decision letter after peer review is shown below.]

Thank you for submitting your work entitled "Lamellar junctions in the endolymphatic sac act as a relief valve to regulate inner ear pressure" for consideration by *eLife*. Your article has been reviewed by three peer reviewers, and the evaluation has been overseen by a Reviewing Editor and a Senior Editor. The following individuals involved in review of your submission have agreed to reveal their identity: Philine Wangemann (Reviewer #1).

Summary:

The reviewers thought the work was interesting and using impressive technology. Although the results are potentially novel with a possible link to human disease, you will see that the reviewers were not completely convinced of the proposed mechanism, i.e. that breaking of the epithelium drives deflation of the endolymphatic sac. A general concern was that the interpretations go beyond the data. In terms of specific comments, the reviewers wanted better validation of the tight function results, and a better demonstration of the presence or absence of basement membrane. Linking opening of the "lamellar junctions" to the deflation process is key to the overall conclusions, but not completely convincing. For example, the videos were not clear to the reviewers in terms of resolving lamella opening. We recognize that this could be technically difficult, but the conclusions should match what can be discerned from the data. The reviewers did not think the Rac1 experiments were informative and thus may want to consider removing them from the manuscript.

Reviewer #1:

This manuscript describes periodic openings of epithelial junctions in the endolymphatic sac of developing zebrafish after the onset of sensory function. The use of high-resolution 3D light microscopy to investigate the dynamic behavior of cells during development opens a new dimension in our understanding of organ development. I would like to make a few comments:

1) The authors observed thinning of epithelial cells and openings of junctions between epithelial cells of the endolymphatic sac and argue that they observed a novel form of junction which they would like to call 'lamellar junction'. I think that the possibility that dynamic behavior of tight junctions resulted in the observed barrier openings has been dismissed too easily and that a new name is therefore not warranted at this time. The authors' main argument for a novel kind of junction appears to rest on the assumption that the thin regions where cell height approaches a thickness of 1 micrometer are too thin to accommodate tight junctions that separate an apical from a basolateral membrane of the epithelium. Accordingly, the authors write: "In wild-type embryos this region often appeared as thin as a single membrane (1.0+/-0.3 micrometer, mean +/- SD) rather than being thick enough to identify the characteristic apical and basal sides of an epithelium". I would like to point out that other very thin epithelia such as alveolar epithelia have no problem to form tight junctions that separate an apical from a basolateral membrane. The thinness of the cells may therefore not be suitable as demission criterion. Before a novel form of an epithelial junction should be claimed and named, it would be necessary to perform more work. It would be necessary to demonstrate the absence of typical electron-dense structures between cell membranes, absence of congregated tight junction proteins such as claudins, occludins and ZO-1 and lack of segregation of marker proteins such as Na+/K^+^ ATPase. Great care would need to be taken to avoid fixation artifacts. Demonstration of uniform expression of marker proteins such as Na+/K^+^ ATPase may provide the necessary evidence to address the thin folds of cell membrane as basal lamellae, a term that implies that these thin folds consist of exclusively of basolateral membrane.

2) Video 5 supports the concept that the ES is continually inflating, however, Video 6 does not. In Video 6, there appears to be at least one set back in volume expansion after time 76:00:00. Observation of Video 6 questions whether the time course in Figure 3D is representative.

3) Comparison of Video 1-4 to Video 8 leads to the impression that cells are swollen in Video 8 which may indicate that AO-LLSM itself or the preparation until the imaging began was more damaging to the tissue than conventional confocal microscopy. Cell swelling including a loss of cytoskeletal integrity and vacuolization may contribute to differences in the apparent elastic properties. Although the centroid-based evaluation of elastic behavior is elegant, the authors are correct to recognize the shortcomings of the technique and to refrain from postulating the presence of multiple cell types distinguished by differences in elastic properties.

4) It is unclear to me what is happening in Video 4? At 76:05.00 a luminal segment appears to get detached before it disappears at 76:17.30. Is the ES fragmenting or are epithelial cells engulfing large vacuoles possibly in a pathologic process?

5) The experiments employing the dominant negative Rac1 mutant are not necessary to arrive at the conclusion that "cytoskeletal regulators are […] used for lamellipodial extension". Nevertheless, the experiments employing Rac1 mutants should be quantified and presented in Volume vs Time graphs. In support of the conclusion, it is important to demonstrate that volumes increase in the Rac1 mutant at least as much if not more than in controls.

Reviewer #2:

The manuscript from Swinburne et al. is an interesting and well-performed piece of work analysing the mechanisms of inflation and deflation of the zebrafish Endolymphatic Sac. The authors combine high resolution life imaging techniques, 3D cell reconstructions, quantitative analysis and EM studies and show for the first time the dynamics of the ES epithelium during inflation and deflation. The authors find lamellar protrusions that might work as a valve and upon breakage contribute to the release of ES pressure and deflation. The authors link the action of Lmx1b gene in this process. The findings are relevant for the understanding the homestasis of endolympha in the inner ear. Disruption of ES function might be involved in relevant inner ear diseases. The work is primarily descriptive at this early stage.

1) One on the main points of the manuscript is that rupturing of the lamellae from ES cells allow the leakage out of fluid from the endolymph to the perilymph, being the main mechanism for deflation of the ES. The leakage is only shown in Figure 2E (75.22) and Video 4. To me, the leakage is not very clear. It seems that the cell membranes are displaced and if there is some leakage is very small and probably cannot account for the large shrinkage of ES volume in deflation. As inflation is due to passage from the main otic vesicle cavity to the ES, the deflation could be also explained by involving a contraction and returning of fluid backwards to the main cavity or to re-absorption by ES cells. Also, it is shown in Figure 1 that at the time of deflation, perilymph can enter the ES. Could this be explained because the ES fluid flows back to the vesicle and then fluid can enter from the perilymph to the ES? This possibility should be addressed by injecting dye in the ES and imaging the flow of the dye. The main point of the paper is to convince that breakage of the epithelium drives deflation, but the real contribution to this event to the deflation is less clear. Major epithelium reorganization takes place during the inflation and deflation that are little explored.

2) In line with this, imaging of F-actin or myosinII during the cycles of inflation and deflation would greatly improve the paper. Is the epithelium contracting during deflation or vice-versa is breakage relevant for releasing tension during deflation?

3) Regarding the Lmx1bb role in inner ear. From the paper, it seems that Lmx1bb is only expressed in the ES which is not and thus the overall expression should be shown. Has the Lmx1bb mutant other defects that should be taking into account? The authors try to make a functional relationship between the phenotype in the ES and the lamellae observed but altogether this data is not too strong. The authors mention that in the Lmx1bb mutant the ES cannot deflate and is continually enlarged. The authors claim that inhibition of deflation, reveals that the diffusion barrier does not break but this is not shown. Moreover, there is not evidences that Lmx1bb is necessary for ES epithelial barrier. From the images Figure 3E it seems that the organization of the ES epithelium is different, but as the authors have performed EM of Lmx1bb mutant ES, they should indicate if lamellae are present. Only one panel is shown of the mutant but should provide a better analysis of the epithelial changes. This data would tell better if indeed loss of the lamellae and increase in the junctional organization of the ES epithelium impairs release of the fluid. In discussion the authors claim that Lmx1b might be regulating the loss of junctional adhesions. Are the separations between cells lost in Lmx1bb mutant? If so how the ES expands? Can the authors image ZO1 localization in the mutants and wild-type during inflation and deflation?

4) The authors mention that the cells stretch upon inflation and perform cell reconstruction with their ACME software. Is the volume of ES cells changing overtime? In Figure 5H changes of ES lumen volume are correlated with changes in ES cell thickness but that should be correlated with mean ES cell volumes. Are ES cells losing fluid during inflation?

The stretching and destretching favours the idea that is not only pressure release by epithelium breakage what triggers deflation but the building up of tension a possible main drive for inflation and deflation of the ES. Again, imaging contractility in wild-type and Lmx1bb mutant might be relevant to discern these two possibilities.

5) I am not too convinced that the data on Rac1 is relevant and informative. If more functional studies on the action of the cytoskeleton is provided, then this data might acquire more relevance, if not I don´t see what adds to the paper.

6) A direct evidence that lamellae opening triggers the leakage is not clear. On one side, the authors mention that they observe separation of lamellae (but this would just be part of the stretching mechanism). On the other side, they indicate that there is leakage out (not clear in Figure 2E). Figure 7 tries to make the link. They show 3D renderings of cells Figure 7A but then indicate that in pink the endolymph is depicted. It is not clear in Figure 7 B and D what is a cell and what is the lumen. The breakage of the epithelium should be indicated in the Figure clearly.

Overall, the authors draw conclusions that lamellae are relevant for opening the epithelium and driving deflation. However, as this data is not straight forward, the lamellae might be relevant for stretching during inflation and contraction drive the inflation- deflation cycles.

Reviewer #3:

This manuscript describes a previously unknown phenomenon of the endolymphatic sac (ES) during zebrafish inner ear development. The authors demonstrated that the ES underwent periodic cycles of inflation and deflation during development. Cells in the ES changed shape dynamically to form lamella projections, which served as a pressure valve to release fluid within the otic vesicle after it has inflated to a certain point. The authors showed that knockout of lmx1b, a gene expressed in the ES, cells within the sac did not undergo cell shape changes and the otic vesicle simply swelled without cycling through the inflation-deflation process. These results are interesting and provide insights into several pathological conditions pertaining to the disruption of fluid homeostasis of the inner ear.

Perhaps, the most puzzling part of the results is the description of perilymph entering into the ES while endolymph was being released under pressure. Have the authors entertained other possibilities or artifacts to account for a rise in the level of fluorescent dyes during deflation? I noticed that during the inflation phase, fluorescent vesicles/droplets were seen in the ES cells, suggesting the possibility that perilymph was being transported into the ear. This observation was not described or discussed in the manuscript. Could the fluorescent dye in the perilymph be transported into the ES cells or endolymph during the inflation phase but somehow the dye concentration increases within the endolymph after the initial endolymph release rather than diffusion of dyes/perilymph into the vesicle at the time of deflation?

Presumably, there is a basement membrane lining the otic vesicle and it is not clear what happens to the basement membrane at the ES and the lamella region. It does not look like there is a basement membrane lining the lamellae from the EM pictures. How would the presence or absence of a basement membrane affect the lamellae opening and closing? Immunostaining for the basement membrane would at least be helpful to determine whether the basement membrane is intact at the endolymphatic duct region but absent or discontinuous at the ES.

1) The ES was described to grow and move towards a more central and medial position during morphogenesis (subsection “The endolymphatic sac exhibits cycles of inflation and deflation”). It would be good to have a video documenting this process and it will complement results shown in Video 1.

2) Based on Video 4 and the summary diagram in Figure 7E, I got the impression that the release valve is at the tip of the ES. However, the light sheet data indicate that the lamella region is highly dynamic suggesting that the pressure relief at the ES could occur at any region of the ES. If correct, then a summary diagram that reflects the entire data is warranted.

3) ES cells in Lmx1b mutant do not undergo lamella formation, which was attributed to the presence of tight junctions. Since tight junctions are normally found in epithelial cells, a more global assessment of tight junctions between wildtype and mutant ES using antibodies to tight-junction proteins or tight-junction proteins coupled to Gfp is warranted.

4) Is Lmx1b expressed in the entire ES? How does the different cellular behaviors described in Figure 5 relate to the expression of lmx1b?

5) Video 4. Please clarify that the labeling of the endolymph at the beginning of the video is the result of a single injection to the otic vesicle rather than two separate injections to the vesicle and ES. It also seems like there could be two events of fluid release from the ES. If so, the text and panels in the figure should be revised accordingly.

[Editors’ note: what now follows is the decision letter after the authors submitted for further consideration.]

Thank you for resubmitting your work entitled "Lamellar projections in the endolymphatic sac act as a relief valve to regulate inner ear pressure" for further consideration at *eLife*. Your revised article has been favorably evaluated by Gary Westbrook (Senior Editor), a Reviewing Editor, and three reviewers.

The reviewers are pleased with the revision, but had a few minor issues that need your attention, as outlined below. Your responses will be monitored by the Senior Editor before acceptance.

Reviewer #1:

This revised manuscript is a huge improvement over the previous submission. The authors have gone beyond their call of duty in addressing the issues raised in the previous submission. The additional results of simultaneous labeling of perilymph and endolymph as well as the identification of the Lmx1b-positive cells in the endolymphatic sac provided clarity and added volume to this manuscript.

Further minor suggestions to improve this manuscript are listed as follow:

1) The new color scheme is very helpful. How about making the color scheme consistent in Figure 3I and J as well? Adding a white arrow in 3I and adding genotype information on Figure 3C, D and E will also help readers to get through this information-packed figure.

2) Figure 3L, Anti-collagen staining pattern in WT seems qualitatively different between 50 and 80 hpf. Is this a reproducible result? If so, the staining pattern of the lmx1b mutant resembles the WT in 50 hpf rather than 80 hpf, which suggests an arrest of development in the mutant. These points should be clarified in the text.

3) Figure 3 legend, 3L, *eLife* probably will require a more specific n number than 6-32.

4) Subsection “Apical and basal myosin during ES inflation cycles”, Do the authors mean video 13 instead of video 11?

5) Discussion section, it may be misleading to say Lmx1a mutant exhibits ES defects like the fish lmx1bb mutant because Lmx1a mouse mutants have no endolymphatic duct or sac. One could say they exhibit similar enlarged ear defects.

6) Introduction, extra brackets.

Reviewer #2:

The authors have improved their excellent manuscript and provided additional data following the reviewers' suggestions. The term "lamellar projections" is indeed a better choice than the initially "lamellar junctions".

Reviewer #3:

The authors have made great efforts to answer the reviewer’s points. They have performed all the experiments requested to address the data that was not clear and that could help to discard other possible scenarios. The experiments have indeed helped to improve substantially the paper, together with major rewriting of the text and incorporation of more videos. The Results section flows very smoothly now and it is to thank that they have also extended the discussion with the three different possibilities of how the cycles of inflation and deflation might be regulated. The role of Lmx1b, as well as of the apical cell remodeling is also more clear in this version.

The paper combines a large set of different techniques, that altogether make a beautiful and also interesting piece of work exploring how hydrostatic pressure might be regulated in the inner ear. This work could be extended to other organs in which fluid homeostasis has to be tightly regulated. The possible role of Lmx1bb in controlling remodeling of apical junctions and lamellae might be relevant in other cellular contexts.

I only suggest that the authors mention in methods section whether the analysis in lmx1bb crispr mutants was done in F1 embryos.

---

## [Author Response]

[Editors' note: the author responses to the re-review follow.]

Reviewer #1:This manuscript describes periodic openings of epithelial junctions in the endolymphatic sac of developing zebrafish after the onset of sensory function. The use of high-resolution 3D light microscopy to investigate the dynamic behavior of cells during development opens a new dimension in our understanding of organ development. I would like to make a few comments:1) The authors observed thinning of epithelial cells and openings of junctions between epithelial cells of the endolymphatic sac and argue that they observed a novel form of junction which they would like to call 'lamellar junction'. I think that the possibility that dynamic behavior of tight junctions resulted in the observed barrier openings has been dismissed too easily and that a new name is therefore not warranted at this time. The authors' main argument for a novel kind of junction appears to rest on the assumption that the thin regions where cell height approaches a thickness of 1 micrometer are too thin to accommodate tight junctions that separate an apical from a basolateral membrane of the epithelium. Accordingly, the authors write: "In wild-type embryos this region often appeared as thin as a single membrane (1.0+/-0.3 micrometer, mean +/- SD) rather than being thick enough to identify the characteristic apical and basal sides of an epithelium". I would like to point out that other very thin epithelia such as alveolar epithelia have no problem to form tight junctions that separate an apical from a basolateral membrane. The thinness of the cells may therefore not be suitable as demission criterion.

We did not mean to suggest that this data is a proof that the ES cells are not polarized. We have reworded the presentation of our observations.

subsection “Lamellar barriers appear to form an ES relief valve”:

“In wild-type embryos, this region often appeared as a thin sheet (1.0 ± 0.3 µm, mean ± SD) rather than thick as seen in the un-inflated wild-type ES or the *lmx1bb* mutants in which two distinct surfaces (apical and basal) were observed (6.2 ± 3.2 µm, the mutant, *n* = 9, is significantly thicker than wild-type, *n* =14, with a Mann-Whitney-Wilcoxon one tailed *p*-value of 4 x 10^-5^).”

We agree with this criticism regarding the potential role of tight junction dynamics and now fully address the potential role for dynamic tight junctions throughout the manuscript.

Subsection “Lamellar barriers appear to form an ES relief valve”:

“In contrast to junctions between thin aveolar cells, which kiss at their tips with tight junctions, the lamellae from ES cells formed large zones of overlap and sometimes bifurcated to form a tongue-and-groove structure (Figure 4B, inset) or interwoven structures (Figure 4C,F, black mesh highlights area of overlap, dotted perimeter indicates full spread). These structures lacked electron-dense signal, indicating that they were unlikely to contain tight junctions between lamellae.”

Subsection “Lamellar barriers appear to form an ES relief valve”:

“We identified electron-dense tight junctions between cells at the tip of the ES. These same cells also had lamellar projections (yellow arrows indicate tight junctions, Figure 4G, 3 serial EM sections, 1.2 μm apart, the diameter of the apical opening is ~1.2μm). The electron-dense signal formed a ring sealing the apical side of these ES cells that was continuous except for an opening that connected fluid from the duct and sac to the exposed lamellae (magenta arrow directed from duct to sac, Figure 4G). Apical junctions were also present in the mutant ES of *lmx1bb^jj410/jj410^* larvae (yellow arrows, Figure 4H,H’). Unlike the wild-type ES, however, the mutant ES appeared to completely lack openings in its apical junctions, as we were unable to identify any in mutants imaged by serial-section scanning EM or transmission EM (4 total mutant ears).”

A revised Figure 7F.

“Throughout our imaging experiments, whether the contrast was produced at plasma membranes, endolymph, or perilymph, we observed what appear to be small pockets of lumen that pinch off from the rest of the ES lumen (see Figures 2D, G, 5G, 6D-E, 7E, and Videos 4, 5, 9, 10, 11, 14). These pockets were not present in the *lmx1bb* mutant, (Figure 3E). The simplest explanation is that fluid gets trapped in the lateral intracellular space as the apical junctions open but before the basal lamellae open or when the basal lamellae close after a deflation event. This is consistent with how the inclusions are often resolved in the next inflation cycle. Incorporating all of our observations, we surmise that increased pressure is managed through a combination of strain that stretches viscoelastic cells in the ES and adhesion distributed across the surface area of dynamic basal lamellae. Excess pressure and volume is then released when small openings form amongst compliant apical junctions of ES cells that transmits the stress of the ear’s hydrostatic pressure to the basal lamellae that hold until they separate to release excess pressure and volume. The elastic properties of the tissue then drive its deflation, basal lamellae reunite, and the homeostatic cycle begins again (Figure 7F).”

Before a novel form of an epithelial junction should be claimed and named, it would be necessary to perform more work. It would be necessary to demonstrate the absence of typical electron-dense structures between cell membranes, absence of congregated tight junction proteins such as claudins, occludins and ZO-1 and lack of segregation of marker proteins such as Na+/K^+^ ATPase. Great care would need to be taken to avoid fixation artifacts. Demonstration of uniform expression of marker proteins such as Na+/K^+^ ATPase may provide the necessary evidence to address the thin folds of cell membrane as basal lamellae, a term that implies that these thin folds consist of exclusively of basolateral membrane.

We find this criticism valid and have now shown that the epithelium is indeed still polarized with apical junction near the lumen containing endolymph and laminin and collagen adjacent to basal lamellae. Additionally, we now refer to the lamellae as lamellar barriers rather than lamellar junctions, so as not to make unsubstantiated interpretations regarding their molecular organization. Indeed, the cell behavior appears more complicated than we initially presented, as apical junctions appear to still be present and potentially function as an additional site of pressure regulation. There are likely three sites of resistance to flow in response to volume and pressure increases: the endolymphatic duct, the apical junctions of the endolymphatic sac, and the dynamic basal lamellae. For various technical reasons, the live dynamics of these three structures were not and still are not simultaneously observable. We have added a discussion on the nature of the cell behavior relative to its polarity and discuss the potential of a multi-tiered relief valve.

We addressed the organization of the ES tissue on subsection “Lamellar barriers appear to form an ES relief valve”:

“To determine the gross organization of the ES tissue, we stained for basal proteins collagen (Figure 3K-L) and laminin (not shown). We found that the basal surface of the ES tissue faces the perilymph outside surrounding the otic vesicle (Figure 3J,K). We also examined localization of the apical marker ZO-1 and found that it stained the ES tissue that faces the endolymph-filled lumen (Figure 3K-L). This apical-basal organization was also present in the *lmx1bb^410/410^*mutant (Figure 3K-L). The absence of local breaks in the mutant ES, the absence of regional thinning in the mutant inflated ES, and the absence of thin basal protrusions suggests that these thin areas may be structurally relevant to ES pressure relief.”

2) Video 5 supports the concept that the ES is continually inflating, however, Video 6 does not. In Video 6, there appears to be at least one set back in volume expansion after time 76:00:00. Observation of Video 6 questions whether the time course in Figure 3D is representative.

The growth rate in Figure 3D does show reduction in the rate of ES growth. We have a follow-up study that is looking into the pressure at which tissue tearing occurs.

3) Comparison of Video 1-4 to Video 8 leads to the impression that cells are swollen in Video 8 which may indicate that AO-LLSM itself or the preparation until the imaging began was more damaging to the tissue than conventional confocal microscopy. Cell swelling including a loss of cytoskeletal integrity and vacuolization may contribute to differences in the apparent elastic properties. Although the centroid-based evaluation of elastic behavior is elegant, the authors are correct to recognize the shortcomings of the technique and to refrain from postulating the presence of multiple cell types distinguished by differences in elastic properties.

It is always a concern that the act of observation perturbs the object being observed. A major design advantage of AO-LLSM is that light energy is distributed across the focal plane. For instance, the total power of the AO-LLSM acquisitions was 11 µW for the 488nm light-sheet and 22 µW for the 560nm light-sheet at the back aperture. These laser powers were lower than those used for the confocal experiments (~50µW for both the 514nm laser and the 594nm laser). Furthermore, the power of the lattice light-sheet is distributed over a much larger volume within the depth of focus (~10^5^ times more volume) of the detection objective, thus leading to gentler excitation that resulted in little to no photo-damage over duration of imaging. The only difference in sample preparation was the absence of a coverslip on top of the embryo during AO-LLSM. The observed membrane curvature is likely the result of better resolution, better signal to noise, and a different (oblique) angle of observation (necessitated to minimize the tissue through which the excitation and emission light passed, in contrast to the shared excitation and emission path of standard laser scanning microscopy). We have observed photo-toxicity in the past during normal confocal imaging, where we used higher power than what we report in this paper to compensate for different transgenics expressing low levels of fluorescent proteins. In these cases, the ES does not inflate when cells appear like soap bubbles and become opaque as they die. Following the AO-LLSM imaging sessions, embryos appeared healthy (tissue was still transparent, and development continued).

In Liu et al., 2018 there are many examples of healthy cells undergoing AO-LLSM imaging.

We briefly address this concern in the text, subsection “Cells stretch during ES inflations”:

“A microscope combining live-cell lattice light-sheets and adaptive optics was built (Liu et al., 2018 in press) and used here to image ES cycles with significantly improved spatial and temporal resolution and reduced photo-damage owing to 2–4 times less laser power being distributed over a large volume (the imaging plane is volumetrically ~10^5^ times larger than the confocal point).”

4) It is unclear to me what is happening in Video 4? At 76:05.00 a luminal segment appears to get detached before it disappears at 76:17.30. Is the ES fragmenting or are epithelial cells engulfing large vacuoles possibly in a pathologic process?

It appears that fluid can get trapped in a lateral space between apical junctions and basal lamellae. We have added more movies exhibiting this phenomenon. We have no reason to think these are pathological, but likely highlight the simultaneous existence of apical adhesion and a basal lamellar barrier. Two barriers likely make the relief valve more robust and we now discuss this in the Discussion section:

“Throughout our imaging experiments, whether the contrast was produced at plasma membranes, endolymph, or perilymph, we observed what appear to be small pockets of lumen that pinch off from the rest of the ES lumen (see Figures 2D, G, 5G, 6D-E, 7E, and Videos 4, 5, 9, 10, 11, 14). These pockets were not present in the *lmx1bb* mutant, (Figure 3E). The simplest explanation is that fluid gets trapped in the lateral intracellular space as the apical junctions open but before the basal lamellae open or when the basal lamellae close after a deflation event. This is consistent with how the inclusions are often resolved in the next inflation cycle. Incorporating all of our observations, we surmise that increased pressure is managed through a combination of strain that stretches viscoelastic cells in the ES and adhesion distributed across the surface area of dynamic basal lamellae. Excess pressure and volume is then released when small openings form amongst compliant apical junctions of ES cells that transmits the stress of the ear’s hydrostatic pressure to the basal lamellae that hold until they separate to release excess pressure and volume. The elastic properties of the tissue then drive its deflation, basal lamellae reunite, and the homeostatic cycle begins again (Figure 7F).”

5) The experiments employing the dominant negative Rac1 mutant are not necessary to arrive at the conclusion that "cytoskeletal regulators are […] used for lamellipodial extension". Nevertheless, the experiments employing Rac1 mutants should be quantified and presented in Volume vs Time graphs. In support of the conclusion, it is important to demonstrate that volumes increase in the Rac1 mutant at least as much if not more than in controls.

The Rac1 dominant negative likely impedes the ability of the basal lamellae to hold hydrostatic pressure. When this occurs depends on when openings are present in the apical junctions. The time of apical junction opening formation seems to vary quite a bit between each wild-type ES based on when labeled endolymph can fill the lateral space (Video 4, Video 5 and Video 14). Therefore, it seems incorrect to assume that the ES would inflate to the same volumes as controls. We have added myosin movies that complement this experiment. Additionally, we have provided more detail to the discussion and the cartoon model in Figure 7F regarding the two barriers to pressure release, apical junctions and basal lamellae.

Reviewer #2:The manuscript from Swinburne et al. is an interesting and well-performed piece of work analysing the mechanisms of inflation and deflation of the zebrafish Endolymphatic Sac. The authors combine high resolution life imaging techniques, 3D cell reconstructions, quantitative analysis and EM studies and show for the first time the dynamics of the ES epithelium during inflation and deflation. The authors find lamellar protrusions that might work as a valve and upon breakage contribute to the release of ES pressure and deflation. The authors link the action of Lmx1b gene in this process. The findings are relevant for the understanding the homestasis of endolympha in the inner ear. Disruption of ES function might be involved in relevant inner ear diseases. The work is primarily descriptive at this early stage.1) One on the main points of the manuscript is that rupturing of the lamellae from ES cells allow the leakage out of fluid from the endolymph to the perilymph, being the main mechanism for deflation of the ES. The leakage is only shown in Figure 2E (75.22) and Video 4. To me, the leakage is not very clear. It seems that the cell membranes are displaced and if there is some leakage is very small and probably cannot account for the large shrinkage of ES volume in deflation.

We have added more images and movies showing leakage. The ES is flanked by blood vessels, that are fenestrated and it is likely that the expelled dye is rapidly taken away and diluted within the vasculature. The rapid and transient nature of the leaked endolymph is likely the result of dilution and advection into the blood stream. See improved Figure 2, Figure 7, Video 4, Video 5 and Video 14.

As inflation is due to passage from the main otic vesicle cavity to the ES, the deflation could be also explained by involving a contraction and returning of fluid backwards to the main cavity or to re-absorption by ES cells. Also, it is shown in Figure 1 that at the time of deflation, perilymph can enter the ES. Could this be explained because the ES fluid flows back to the vesicle and then fluid can enter from the perilymph to the ES? This possibility should be addressed by injecting dye in the ES and imaging the flow of the dye. The main point of the paper is to convince that breakage of the epithelium drives deflation, but the real contribution to this event to the deflation is less clear. Major epithelium reorganization takes place during the inflation and deflation that are little explored.

We have now added a representative time-lapse movie where both the endolymph and perilymph are labeled, in two different colors. These movies clearly show the expulsion of fluid from the ES into the perilymphatic space (as in the prior Figure 2E and Video 4). We show more instances of the fluid movement and have made an effort to clarify both the presentation in the figure as well as the movies.

Additionally, we have now examined the localization of myosin throughout the inflation and deflation cycle, and there is no evidence for an active pumping behavior that has greater contractility than that present during inflation. Additionally, puncturing of wild-type and mutant otic vesicles is sufficient to cause the elastic collapse of the ES. We now discuss how constant tension could contribute to both the resistance to inflation as well as elastic collapse during deflation.

Subsection “Apical and basal myosin during ES inflation cycles”:

“Cells and tissues modulate tension for crawling, retracting, bending, constricting, rounding-up, and resisting stretch by organizing an extensive variety of complexes of actin filaments, actin binding proteins, and myosin motors (Grill, 2011; Munjal and Lecuit, 2014; Rafelski and Theriot, 2004; Stewart et al., 2011). Myosin distribution correlates strongly with contractile forces (Fournier et al., 2010). Therefore, to begin to determine how myosin and actin might contribute to tension during ES inflation and deflation, we imaged myosin and actin using non-muscle myosin light chain fused to eGFP and utrophin, an actin binding protein, fused to mCherry (confocal microscopy, Figure 6H, Video 12). Myosin localized to the apical domains of ES cells as well as to dynamic puncta at the basal membrane throughout both inflation and deflation of the ES. During ES inflation, contraction of apical myosin likely counteracts the stress of hydrostatic pressure and maintains the integrity of apical junctions (Rauzi et al., 2010). Tension from apical myosin, strain from hydrostatic pressure, and the adhesion strength between junctional protein complexes likely determines the elastic limit of apical junctions. The crossing of this elastic limit may cause the small apical openings observed in Figure 4G, which ultimately communicates the otic vesicle’s hydrostatic pressure to the lamellar protrusions. The dynamic spots of basal myosin could indicate two possible activities: contraction to retract lamellar protrusions as observed during cell migration or contraction at focal adhesions. Utrophin localized strongest at apical domains of ES cells (Figure 6H), but unfortunately bleached too quickly to generate any meaningful insights into the potential of actin localization dynamics. It had been shown that utrophin based actin highlighters do not localize to actin within lamellae (Belin et al., 2014), which explains the relatively low amounts of basal utrophin signal. In contrast, fluorescent phalloidin labels actin filament enrichment at both apical and basal domains of ES cells (not shown). Our observations of consistent apical localization of myosin throughout ES cycling suggests that the same contractility both counteracts hydrostatic pressure during inflation and drives deflation when hydrostatic pressure drops as the relief valve is opened. We did not observe any sudden changes in myosin localization or behavior that would indicate regulation for either inflation or deflation. Analysis of myosin localization in *lmx1bb* crispants (CRISPR-Cas9 knockout embryos) revealed similar apical localization and dynamic basal puncta (Figure 6I, Video 12, representative of 2 time-lapses). To determine whether the mutant ES tissue has elastic material properties, we punctured the otic vesicle with a tungsten needle and found that the distended ES rapidly collapsed (5/5 mutant puncture experiments caused ES collapse, Figure 6K). Global inhibition of actin or myosin with cytochalasin D or (S)-nitro-blebbistatin correlated with ES deflation. However, it is currently impossible to determine whether this is owed to hydrostatic pressure not being maintained by the otic vesicle epithelium.”

Discussion section

“Additionally, myosin appears to act consistently throughout the ES cycles likely providing a constant apical tension that both resists pressure-induced stretch during inflation and drives collapse during deflation by contributing to tissue elasticity. This constant activity precludes the need for sensors or signals to tell the valve when to inflate, when to open, and when to collapse. Furthermore, dynamic basal puncta of myosin may provide integrity to adhesion between basal lamellae with added tension (Ingber, 1997).”

2) In line with this, imaging of F-actin or myosinII during the cycles of inflation and deflation would greatly improve the paper. Is the epithelium contracting during deflation or viceversa is breakage relevant for releasing tension during deflation?

We agree that the epithelium is not passive in the process of inflation and deflation. We have imaged both utrophin and myosin dynamics. Myosin is present within the apical domain of ES cells throughout both inflation and deflation. Basal myosin is dynamic during both inflation and deflation. All our data suggests that contractility that resists the stress of hydrostatic pressure during inflation is used for the elastic behavior of the ES during deflation. See the changes to the text highlighted above. A full study of the role of actin and myosin’s roles in the ES is beyond the scope of this paper and is an ongoing effort.

3) Regarding the Lmx1bb role in inner ear. From the paper, it seems that Lmx1bb is only expressed in the ES which is not and thus the overall expression should be shown. Has the Lmx1bb mutant other defects that should be taking into account?

We now address expression of lmx1bb in other tissues.

Subsection “Lmx1bb is essential for ES deflation”:

“Earlier in the development of the otic vesicle, *lmx1bb* is expressed in portions of the nascent semicircular canals and sensory patches, regions of the inner ear that also exhibit abnormal development in the mutant (Obholzer et al., 2012; Schibler and Malicki, 2007). There is no precedent for ES development being dependent on those portions of the otic vesicles and there are many mutants with similar SCC or sensory defects that do not have ES phenotypes (Fekete, 1999; Malicki et al., 1996; Whitfield et al., 1996).”

The authors try to make a functional relationship between the phenotype in the ES and the lamellae observed but altogether this data is not too strong. The authors mention that in the Lmx1bb mutant the ES cannot deflate and is continually enlarged. The authors claim that inhibition of deflation, reveals that the diffusion barrier does not break but this is not shown.

The evidence that the diffusion barrier does not break in the mutant was presented in Figure 3D and supplemental figure for Figure 3—figure supplement 1. On the secondary axis we plotted the leak in fluorescence of perilymph into the ES lumen. Unlike in the wild-type, there was never any leak-in fluorescence. Additionally, we performed new time-lapse experiments where the endolymph was labeled in mutants. We also improved the writing to emphasize the evidence and clarify our interpretation.

Subsection “Lmx1bb is essential for ES deflation”:

“As in the wild-type analysis, we quantified the presence of perilymph leaking into the ES lumen (secondary axes of Figure 3D and Figure 3—figure supplement 1A). In the mutant, however, we never observed perilymph entering the ES. Additionally, we imaged mutants where the endolymph was labeled and did not observe leakage out of the distended ES epithelium (Figure 3E). These findings suggest that the epithelial diffusion barrier remains intact in the mutant ES.”

Moreover, there is not evidences that Lmx1bb is necessary for ES epithelial barrier. From the images Figure 3E it seems that the organization of the ES epithelium is different, but as the authors have performed EM of Lmx1bb mutant ES, they should indicate if lamellae are present. Only one panel is shown of the mutant but should provide a better analysis of the epithelial changes. This data would tell better if indeed loss of the lamellae and increase in the junctional organization of the ES epithelium impairs release of the fluid. In discussion the authors claim that Lmx1b might be regulating the loss of junctional adhesions. Are the separations between cells lost in Lmx1bb mutant? If so how the ES expands? Can the authors image ZO1 localization in the mutants and wild-type during inflation and deflation?

We do not state or find any evidence that Lmx1bb is necessary for the ES barrier. Our findings support a role for lmx1bb in weakening or removing apical junctions to create openings that connect the duct to the basal lamellae. We have performed more EM of the mutants and now present images indicating that tight junctions are present in the wild-type and mutant. However, in the wild-type, there are small apical openings that are not present in the mutant. We have added more EM images and whole mount stains. Unfortunately, it is not currently possible to live-image tight junction proteins in the mutants and wild-type. Ubiquitous overexpression of ZO1-GFP sickens embryos. We are working hard to develop technology to perform these experiments, but this is likely a long-term endeavor.

Subsection “Lamellar barriers appear to form an ES relief valve”:

Sparse mosaic labeling of cells in the wild-type further confirmed the presence of thin basal protrusions in the ES, as well as diversity in their organization (Figure 3I). Similar labeling in the mutant revealed the absence of thin basal protrusions in the ES (Figure 3J).

The following additions have been made to the Results section:

“We identified electron-dense tight junctions between cells at the tip of the ES. These same cells also had lamellar projections (yellow arrows indicate tight junctions, Figure 4G, 3 serial EM sections, 1.2 μm apart, the diameter of the apical opening is ~1.2μm). The electron-dense signal formed a ring sealing the apical side of these ES cells that was continuous except for an opening that connected fluid from the duct and sac to the exposed lamellae (magenta arrow directed from duct to sac, Figure 4G). Apical junctions were also present in the mutant ES of *lmx1bb^jj410/jj410^* larvae (yellow arrows, Figure 4H,H’). Unlike the wild-type ES, however, the mutant ES appeared to completely lack openings in its apical junctions, as we were unable to identify any in mutants imaged by serial-section scanning EM or transmission EM (4 total mutant ears). Consistent with the absence of thin protrusions in the sparsely labeled mutants (Figure 3J), we could not find long lamellar projections on the basal side of the mutant ES (Figure 4I). We were unable to identify apical openings in the wild-type ZO-1 immunostains (Figure 3L). This could be due to difficulty resolving small openings (~1μm diameter) or to ZO-1 remaining present at the openings such that the apical barrier could reform after ES deflation to hold hydrostatic pressure. These data suggest that *lmx1bb*-dependent activity is necessary for openings to form in the apical junctions of ES tissue and for ES cells to extend thin basal protrusions.”

4) The authors mention that the cells stretch upon inflation and perform cell reconstruction with their ACME software. Is the volume of ES cells changing overtime? In Figure 5H changes of ES lumen volume are correlated with changes in ES cell thickness but that should be correlated with mean ES cell volumes. Are ES cells losing fluid during inflation?

We do not see any evidence of gross loss of fluid by the ES cells. While cell volume measurements from microscopy data exist in the literature, they are generally flawed in that anisotropy in the point spread function makes volume measurements very susceptible to systematic error depending on how anisotropic cells are oriented relative to the imaging plane. Together, our imaging and segmentation are not accurate enough to detect subtle changes in cell volume (~10%).

The stretching and destretching favours the idea that is not only pressure release by epithelium breakage what triggers deflation but the building up of tension a possible main drive for inflation and deflation of the ES. Again, imaging contractility in wild-type and Lmx1bb mutant might be relevant to discern these two possibilities.

We agree that tension is likely critical for both deflation and inflation. This perspective complements our original focus on hydrostatic pressure, as pressure creates the stress that is balanced by tension. Indeed, ablation and puncturing experiments demonstrate that the tissue is elastic. This is most likely because of actomyosin contractility. Imaging myosin through the inflation and deflation cycles shows a constant presence at both apical and basal domains of the ES cells. We now address this perspective in greater detail in the Discussion section:

“Upon first observing the inflation and deflation cycles of the ES, we postulated that the underlying mechanism could either be a response to organ-wide hydrostatic pressure within the otic vesicle, a local tissue behavior where the ES inflates with fluid from the perilymph, or a local tissue behavior where cells in the ES periodically coordinate their movements to expand the ES volume. Our ablation experiments indicated that pressure is transmitted to the ES from the OV and that the tissue is elastic because it collapses quickly when the stress from hydrostatic pressure is removed (Figure 2E-F). The second and third models lack support: dye tracing shows that during inflation the ES is filled with endolymph from the otic vesicle, while perilymph only enters the ES during deflation when the epithelium is open. We did not observe coordinated cell movements within the ES other than stretching. Additionally, myosin appears to act consistently throughout the ES cycles likely providing a constant apical tension that both resists pressure-induced stretch during inflation and drives collapse during deflation by contributing to tissue elasticity. This constant activity precludes the need for sensors or signals to tell the valve when to inflate, when to open, and when to collapse. Furthermore, dynamic basal puncta of myosin may provide integrity to adhesion between basal lamellae with added tension (Ingber, 1997). Throughout our imaging experiments, whether the contrast was produced at plasma membranes, endolymph, or perilymph, we observed what appear to be small pockets of lumen that pinch off from the rest of the ES lumen (see Figures 2D, G, 5G, 6D-E, 7E, and Videos 4, 5, 9, 10, 11, 14). These pockets were not present in the *lmx1bb* mutant, (Figure 3E). The simplest explanation is that fluid gets trapped in the lateral intracellular space as the apical junctions open but before the basal lamellae open or when the basal lamellae close after a deflation event. This is consistent with how the inclusions are often resolved in the next inflation cycle. Incorporating all of our observations, we surmise that increased pressure is managed through a combination of strain that stretches viscoelastic cells in the ES and adhesion distributed across the surface area of dynamic basal lamellae. Excess pressure and volume is then released when small openings form amongst compliant apical junctions of ES cells that transmits the stress of the ear’s hydrostatic pressure to the basal lamellae that hold until they separate to release excess pressure and volume. The elastic properties of the tissue then drive its deflation, basal lamellae reunite, and the homeostatic cycle begins again (Figure 7F).”

5) I am not too convinced that the data on Rac1 is relevant and informative. If more functional studies on the action of the cytoskeleton is provided, then this data might acquire more relevance, if not I don´t see what adds to the paper.

Good suggestion, we have added actin and myosin experiments that complement the Rac1 experiments.

6) A direct evidence that lamellae opening triggers the leakage is not clear. On one side, the authors mention that they observe separation of lamellae (but this would just be part of the stretching mechanism). On the other side, they indicate that there is leakage out (not clear in Figure 2E). Figure 7 tries to make the link. They show 3D renderings of cells Figure 7A but then indicate that in pink the endolymph is depicted. It is not clear in Figure 7 B and D what is a cell and what is the lumen. The breakage of the epithelium should be indicated in the Figure clearly.

These openings are small (as seen in Figure 4D) and are indicated with yellow arrows. We have added additional labels for endolymph and perilymph. Additionally, we have added data from new time-lapses using labeled endolymph, with higher time resolution, that make the narrow paths of endolymph release more clear (Figure 7E, Video 14).

Overall, the authors draw conclusions that lamellae are relevant for opening the epithelium and driving deflation. However, as this data is not straight forward, the lamellae might be relevant for stretching during inflation and contraction drive the inflation- deflation cycles.

We now discuss this more in the Discussion section. We also discuss fluid being trapped between the apical junctions and basal lamellae (Figure 2, 7, Video 4, Video 5 and Video 14), which indicates that both the apical junction and basal lamellae are barriers to pressure release.

Reviewer #3:This manuscript describes a previously unknown phenomenon of the endolymphatic sac (ES) during zebrafish inner ear development. The authors demonstrated that the ES underwent periodic cycles of inflation and deflation during development. Cells in the ES changed shape dynamically to form lamella projections, which served as a pressure valve to release fluid within the otic vesicle after it has inflated to a certain point. The authors showed that knockout of lmx1b, a gene expressed in the ES, cells within the sac did not undergo cell shape changes and the otic vesicle simply swelled without cycling through the inflation-deflation process. These results are interesting and provide insights into several pathological conditions pertaining to the disruption of fluid homeostasis of the inner ear.Perhaps, the most puzzling part of the results is the description of perilymph entering into the ES while endolymph was being released under pressure. Have the authors entertained other possibilities or artifacts to account for a rise in the level of fluorescent dyes during deflation? I noticed that during the inflation phase, fluorescent vesicles/droplets were seen in the ES cells, suggesting the possibility that perilymph was being transported into the ear. This observation was not described or discussed in the manuscript. Could the fluorescent dye in the perilymph be transported into the ES cells or endolymph during the inflation phase but somehow the dye concentration increases within the endolymph after the initial endolymph release rather than diffusion of dyes/perilymph into the vesicle at the time of deflation?

We never observe rapid loss of the cytoplasmic signal of perilymph, just the slow accumulation over time. After labeling perilymph, the first deflation events coincide with little to no accumulation of perilymph dye within the cells, yet still the dye leaked into the endolymph. We modeled the respective roles of advective as well as diffusive flow of the perilymph dye and indeed, the dye could diffuse through a small opening to enter the ES lumen during deflation.

We have also performed new experiments where both the endolymph and perilymph are labeled with different colored fluorescent dyes. These data, as well as an improved discussion of the path of pressure relief, address these concerns.

Figure 2G, Video 5, subsection “Loss of the epithelial barrier is sufficient for ES deflation”:

“To determine whether endolymph efflux occurs at the same time as perilymph influx, we simultaneously labeled the two fluids with different colored dyes and imaged their localization with higher temporal resolution (Figure 2G, Video 5). We found a fluttering behavior underlying deflation in which endolymph leaked out (62:41:20), perilymph leaked in (62:55:20), leaked-in perilymph flushed out with more endolymph efflux (62:55:40), followed by complete deflation (63:12:00).”

Presumably, there is a basement membrane lining the otic vesicle and it is not clear what happens to the basement membrane at the ES and the lamella region. It does not look like there is a basement membrane lining the lamellae from the EM pictures. How would the presence or absence of a basement membrane affect the lamellae opening and closing? Immunostaining for the basement membrane would at least be helpful to determine whether the basement membrane is intact at the endolymphatic duct region but absent or discontinuous at the ES.

We have now immunostained for laminin and collagen and found that they are present but potentially at a reduced amount at the tip of the ES, consistent with the EM micrographs.

1) The ES was described to grow and move towards a more central and medial position during morphogenesis (subsection “The endolymphatic sac exhibits cycles of inflation and deflation”). It would be good to have a video documenting this process and it will complement results shown in Video 1.

We have taken this out of the description as it distracts from the main findings.

2) Based on Video 4 and the summary diagram in Figure 7E, I got the impression that the release valve is at the tip of the ES. However, the light sheet data indicate that the lamella region is highly dynamic suggesting that the pressure relief at the ES could occur at any region of the ES. If correct, then a summary diagram that reflects the entire data is warranted.

Yes, this appears to be a common feature that different lamellae are responsible for different pressure relief cycles. While the initial diagram did not reflect this complexity, we now present this in Figure 7F as well as present the implications of this observation in the discussion, as mentioned above.

3) ES cells in Lmx1b mutant do not undergo lamella formation, which was attributed to the presence of tight junctions. Since tight junctions are normally found in epithelial cells, a more global assessment of tight junctions between wildtype and mutant ES using antibodies to tight-junction proteins or tight-junction proteins coupled to Gfp is warranted.

We have performed better assessment of the tight junctions and find them present in the wild-type and mutant ES. Live imaging of tight-junction proteins in the ES is not currently possible; we look forward to doing these experiments in the future.

4) Is Lmx1b expressed in the entire ES? How does the different cellular behaviors described in Figure 5 relate to the expression of lmx1b?

Lmx1bb appears to be expressed in a subset of the ES. A more thorough analysis of the cell types and molecular states of the ES cells is part of a larger ongoing effort and beyond the scope of this manuscript.

5) Video 4. Please clarify that the labeling of the endolymph at the beginning of the video is the result of a single injection to the otic vesicle rather than two separate injections to the vesicle and ES. It also seems like there could be two events of fluid release from the ES. If so, the text and panels in the figure should be revised accordingly.

We have clarified the video presentation and its discussion in the text.

Subsection “Loss of the epithelial barrier is sufficient for ES deflation”:

To assess whether ES inflations occur by transmission of endolymph pressure between the otic vesicle and the ES, we performed a single injection of a small volume of solution containing 3 kDa fluorescent dextran into the otic vesicle and followed its movement during inflation and deflation.

This has been clarified in the legend of Figure 2:

(B) Time points of individual sagittal slices of raw data from 3D time course (endolymph labeled yellow by single dye injection into otic vesicle, membrane citrine in cyan).

[Editors' note: the author responses to the re-review follow.]

Reviewer #1:This revised manuscript is a huge improvement over the previous submission. The authors have gone beyond their call of duty in addressing the issues raised in the previous submission. The additional results of simultaneous labeling of perilymph and endolymph as well as the identification of the Lmx1b-positive cells in the endolymphatic sac provided clarity and added volume to this manuscript.Further minor suggestions to improve this manuscript are listed as follow:1) The new color scheme is very helpful. How about making the color scheme consistent in Figure 3I and J as well? Adding a white arrow in 3I and adding genotype information on Figure 3C, D and E will also help readers to get through this information-packed figure.

We are glad that the improved color scheme helps. In our imaging experiments we reused sources of contrast in different combinations. As such, there did not exist a color scheme that provides contrast for visualization in all combinations. We did continue with our color scheme of using green for membrane-citrine in Figure 3I and J. We made an exception regarding the color scheme in Figure 3 I and J because, what maximized contrast for other experiments, i.e. green for membrane-citrine and yellow for membrane-mCherry2, would have made it difficult to discern the two channels. One of the main reasons for presenting information with color is to provide contrast between different objects. There is not enough contrast between green and yellow—therefore we chose to maximize contrast with green and magenta. We hope that clear labeling helps to prevent any confusion.

We have added genotype information and white arrows to Figure 3I.

2) Figure 3L, Anti-collagen staining pattern in WT seems qualitatively different between 50 and 80 hpf. Is this a reproducible result? If so, the staining pattern of the lmx1b mutant resembles the WT in 50 hpf rather than 80 hpf, which suggests an arrest of development in the mutant. These points should be clarified in the text.

We clarified these points in the text, in subsection “Lamellar barriers appear to form an ES relief valve”:

3) Figure 3 legend, 3L, eLife probably will require a more specific n number than 6-32.

We now indicate the specific n number for each stain presented in Figure 3L in the figure legend.

4) Subsection “Apical and basal myosin during ES inflation cycles”, Do the authors mean video 13 instead of video 11?

Yes, this has been corrected. Thanks!

5) Discussion section, it may be misleading to say Lmx1a mutant exhibits ES defects like the fish lmx1bb mutant because Lmx1a mouse mutants have no endolymphatic duct or sac. One could say they exhibit similar enlarged ear defects.

We have clarified the text. The work in Koo et al., indicates that the endolymphatic duct is not discernable by paint fill in end-point analyses. It is possible that a distended lumen would cause a nascent duct and sac to lapse into wall of the otocyst but this will await further investigation.

6) Introduction, extra brackets.Reviewer #3:The authors have made great efforts to answer the reviewer’s points. They have performed all the experiments requested to address the data that was not clear and that could help to discard other possible scenarios. The experiments have indeed helped to improve substantially the paper, together with major rewriting of the text and incorporation of more videos. The Results section flows very smoothly now, and it is to thank that they have also extended the discussion with the three different possibilities of how the cycles of inflation and deflation might be regulated. The role of Lmx1b, as well as of the apical cell remodeling is also more clear in this version.The paper combines a large set of different techniques, that altogether make a beautiful and also interesting piece of work exploring how hydrostatic pressure might be regulated in the inner ear. This work could be extended to other organs in which fluid homeostasis has to be tightly regulated. The possible role of Lmx1bb in controlling remodeling of apical junctions and lamellae might be relevant in other cellular contexts.I only suggest that the authors mention in Materials and methods section whether the analysis in lmx1bb crispr mutants was done in F1 embryos.

We have clarified the text in subsection “In situ and fluorescent immunostains”. We would like to point out that F0 crispants were only used for the myosin-EGFP analysis to avoid a ~3-month delay due to crossing. The other experiments in the paper were done with the established germ-line allele *lmx1bb^jj410^*.